# MetaCURL: Non-stationary Concave Utility Reinforcement Learning

**Bianca Marin Moreno**
Inria*
EDF R&D†

**Margaux Brégère**
Sorbonne Université‡
EDF R&D†

**Pierre Gaillard**
Inria*

**Nadia Oudjane**
EDF R&D†

## Abstract

We explore online learning in episodic Markov decision processes on non-stationary environments (changing losses and probability transitions). Our focus is on the Concave Utility Reinforcement Learning problem (CURL), an extension of classical RL for handling convex performance criteria in state-action distributions induced by agent policies. While various machine learning problems can be written as CURL, its non-linearity invalidates traditional Bellman equations. Despite recent solutions to classical CURL, none address non-stationary MDPs. This paper introduces MetaCURL, the first CURL algorithm for non-stationary MDPs. It employs a meta-algorithm running multiple black-box algorithms instances over different intervals, aggregating outputs via a sleeping expert framework. The key hurdle is partial information due to MDP uncertainty. Under partial information on the probability transitions (uncertainty and non-stationarity coming only from external noise, independent of agent state-action pairs), the algorithm achieves optimal dynamic regret without prior knowledge of MDP changes. Unlike approaches for RL, MetaCURL handles adversarial losses. We believe our approach for managing non-stationarity with experts can be of interest to the RL community.

## 1 Introduction

We consider the task of learning in an episodic Markov decision process (MDP) with a finite state space $\mathcal{X}$, a finite action space $\mathcal{A}$, episodes of length $N$, and a probability transition kernel $p := (p_n)_{n \in [N]}$ such that for all $(x, a) \in \mathcal{X} \times \mathcal{A}$, $p_n(\cdot|x, a) \in \mathcal{S}_{\mathcal{X}}$. For any finite set $\mathcal{B}$, we denote by $\mathcal{S}_{\mathcal{B}}$ the simplex induced by this set, and by $|\mathcal{B}|$ its cardinality. For all $d \in \mathbb{N}$ we let $[d] := \{1, \ldots, d\}$. At each time step $n$, an agent in state $x_n$ chooses an action $a_n \sim \pi_n(\cdot|x_n)$ by means of a policy, and moves to the next state $x_{n+1} \sim p_{n+1}(\cdot|x_n, a_n)$, inducing a state-action distribution sequence $\mu^{\pi,p} := (\mu_n^{\pi,p})_{n \in [N]}$, where $\mu_n^{\pi,p} \in \mathcal{S}_{\mathcal{X} \times \mathcal{A}}$ for all $n \in [N]$.

In many applications of learning in episodic MDPs, an agent aims at finding an optimal policy $\pi$ maximizing/minimizing a concave/convex function $F$ of its state-action distribution, known as the Concave Utility Reinforcement Learning (CURL) problem:

$$\min_{\pi \in (\mathcal{S}_{\mathcal{A}})^{\mathcal{X} \times N}} F(\mu^{\pi,p}). \tag{1}$$

CURL extends reinforcement learning (RL) from linear to convex losses. Many machine learning problems can be written in the CURL setting, including: RL, where for a loss function $\ell$, $F(\mu^{\pi,p}) = \langle \ell, \mu^{\pi,p} \rangle$; pure RL exploration [28], where $F(\mu^{\pi,p}) = \langle \mu^{\pi,p}, \log(\mu^{\pi,p}) \rangle$; imitation learning [26, 35] and apprenticeship learning [55, 1], where $F(\mu^{\pi,p}) = D_g(\mu^{\pi,p}, \mu^*)$, with $D_g$ representing a Bregman

38th Conference on Neural Information Processing Systems (NeurIPS 2024).

*Univ. Grenoble Alpes, Inria, CNRS, Grenoble INP, LJK, 38000 Grenoble, France.

†EDF Lab, 7 bd Gaspard Monge, 91120 Palaiseau, France

‡Sorbonne Université LPSM, Paris, France

divergence induced by a function $g$ and $\mu^*$ being a behavior to be imitated; certain instances of mean-field control [7], where $F(\mu^{\pi,p}) = \langle \ell(\mu^{\pi,p}), \mu^{\pi,p} \rangle$; mean-field games with potential rewards [34]; among others. The CURL problem alters the additive structure inherent in standard RL, invalidating the classical Bellman equations, requiring the development of new algorithms.

Most of existing works on CURL focus on stationary environments [28, 57, 58, 5, 56, 25, 12, 11], where both the objective function $F$ and the probability transition kernel $p$ remain the same across episodes. However, in practical scenarios, environments are rarely stationary. The work of [39] is the first to address online CURL with objective functions that can change arbitrarily between episodes, also known as adversarial losses [19]. However, their work assumes stationary probability kernels and presents results in terms of static regret (performance comparable to an optimal policy). In non-stationary scenarios, it is more relevant to minimize dynamic regret—the gap between the learner's total loss and that of any policy sequence (see Eq. (5) for formal definition). In this work we address this problem by introducing the first algorithm for CURL handling adversarial objective functions and non-stationary probability transitions, achieving near-optimal dynamic regret.

**High-level idea.** Our approach, called MetaCURL, draws inspiration from the online learning literature. In online learning [9], non-stationarity is often managed by running multiple black-box algorithm instances from various starting points and dynamically selecting the best performer using an "expert" algorithm. This strategy has demonstrated effectiveness in settings with complete information [29, 59, 47, 33]. With MetaCURL, we extend this concept to decision-making in MDPs. Unlike classical online learning, the main challenge faced is uncertainty. We assume that the probability transition kernel in each episode has a known deterministic structure but is affected by an external noise with unknown distribution, placing us in a setting with only partial information (see Section 2 for more details). The learner is then unable to observe the agent's loss under policies other than the one played.

MetaCURL is a general algorithm that can be applied with any black-box algorithm with low dynamic regret in near-stationary environments. CURL approaches suitable as black-boxes rely on parametric algorithms that would require prior knowledge of the MDP changes to tune their learning rate. MetaCURL also addresses this challenge by simultaneously running multiple learning rates and weighting them in direct proportion to their empirical performance. MetaCURL achieves optimal regret of order $\tilde{O}\big(\sqrt{\Delta^{\pi^*} T} + \min\{\sqrt{\Delta^p_\infty T},\ T^{2/3}(\Delta^p)^{1/3}\}\big)$, where $\Delta^p_\infty$ and $\Delta^p$ represent the frequency and magnitude of changes of the probability transition kernel respectively, and $\Delta^{\pi^*}$ is the magnitude of changes of the policy sequence we compare ourselves with in dynamic regret (see Eqs. (6) and (7) for formal definitions). MetaCURL does not require previous knowledge of the degree of non-stationarity of the environment, and can handle adversarial losses. To ensure completeness, we show that Greedy MD-CURL from [39] fulfills the requirements to serve as a black-box algorithm. This is the first dynamic regret analysis for a CURL approach.

**Comparisons.** Without literature on non-stationary CURL, we review non-stationary RL approaches. Most methods [24, 13, 45, 17, 20, 40, 21] typically rely on prior knowledge of the MDP's non-stationarity degree, while MetaCURL does not. Let $\Delta^l_\infty$ and $\Delta^l$ represent the frequency and magnitude of change in the RL loss function, respectively[1]. Recently, [54] achieved a regret of $\tilde{O}\big(\min\{\sqrt{(\Delta^p_\infty + \Delta^l_\infty)T},\ T^{2/3}(\Delta^p + \Delta^l)^{1/3}\}\big)$, a near-optimal result as demonstrated by [40], without requiring prior knowledge of the environment's variation. However, this regret bound is tied to changes in loss functions, making it ineffective against adversarial losses. In contrast, rather than

Table 1: Comparisons of our results with the state-of-the-art in non-stationary RL. Here, $\Delta^p_\infty$, $\Delta^p$ and $\Delta^{\pi^*}$ are defined in (6) and (7); and $\Delta^l_\infty$ and $\Delta^l$ measure the RL loss function variations[1]. We introduce $D_T(\Delta_\infty, \Delta) := \min\{\sqrt{\Delta_\infty T},\ T^{2/3}\Delta^{1/3}\}$.

| Algorithm | Dynamic Regret in $\tilde{O}$ | RL | CURL | Adv. losses | No prior knowledge | Exploration |
|---|---|---|---|---|---|---|
| MetaCURL (ours) | $D_T(\Delta^p_\infty, \Delta^p) + \sqrt{\Delta^{\pi^*} T}$ | ✓ | ✓ | ✓ | ✓ | ✗ |
| SoTA in RL [54] | $D_T(\Delta^p_\infty + \Delta^l_\infty, \Delta^p + \Delta^l)$ | ✓ | ✗ | ✗ | ✓ | ✓ |

---

[1]$\Delta^l := 1 + \sum_{t=1}^{T-1} \Delta^l_t$ and $\Delta^l_\infty := 1 + \sum_{t=1}^{T-1} \mathbb{1}\{\Delta^l_t \neq \Delta^l_{t+1}\}$, where $\Delta^l_t := \sum_{n=1}^N \max_{x,a} |\ell^t_n(x,a) - \ell^{t+1}_n(x,a)|$ and $\ell^t_n(x,a)$ is the expected loss suffered by choosing action $a$ in state $x$ at step $n$ of round $t$.

depending on the magnitude of variation of the loss function, MetaCURL's bound depends on the magnitude of variation of the policy sequence we use for comparison in dynamic regret. This allows it to handle adversarial losses, and to compare against policies with a more favorable bias-variance trade-off, which may not align with the optimal policies for each loss. In addition, we improve this dependency by paying it as $\sqrt{\Delta^{\pi^*}T}$ instead of $(\Delta^{\pi^*})^{1/3}T^{2/3}$. We summarize comparisons in Table 1.

**Other related works.** The studies by [43, 42] examine the difference between optimizing the objective over infinite trials and the expectation of the objective over a single trial, challenging the traditional CURL formulation in Eq. (1). Here, we retain the classic formulation to align with existing CURL research. Other works on RL with nonlinear objective functions are [46, 16] focusing on rewards over trajectories rather than individual states. In addition to non-stationarity, there is a series of works on RL with adversarial losses but *stationary* probability transitions, with results only on static regret [48, 30, 18, 50, 32, 14]. Another line of research is known as corruption-robust RL. It differs from non-stationary MDPs in that it assumes a ground-truth MDP and measures adversary malice by the degree of ground-truth corruption [31, 38, 10, 60, 53].

**Contributions.** We resume our main contributions below:

- We introduce MetaCURL, the first algorithm for non-stationary CURL. Under the framework described in Section 2, MetaCURL achieves the optimal dynamic regret bound of order $\tilde{O}\big(\sqrt{\Delta^{\pi^*}T} + \min\{\sqrt{\Delta^p_\infty T}, T^{2/3}(\Delta^p)^{1/3}\}\big)$, without requiring previous knowledge of the degree of non-stationarity of the MDP. MetaCURL handles full adversarial losses and improves the dependency of the regret on the total variation of policies. MetaCURL is the first adaptation of Learning with Expert Advice (LEA) to deal with uncertainty in non-stationary MDPs.
- We also establish the first dynamic regret upper bound for an online CURL algorithm in a nearly stationary environment, which can serve as a black-box routine for MetaCURL.

**Notations.** Let $\|\cdot\|_1$ be the $L_1$ norm, and for all $v := (v_n)_{n\in[N]}$, such that $v_n \in \mathbb{R}^{\mathcal{X}\times\mathcal{A}}$ we define $\|v\|_{\infty,1} := \sup_{1\le n\le N}\|v_n\|_1$.

## 2   General framework: non-stationary CURL

When an agent plays a policy $\pi := (\pi_n)_{n\in[N]}$ in an episodic MDP with probability transition $p$, it induces a state-action distribution sequence (also called the occupancy-measure [61]), which we denote by $\mu^{\pi,p} := (\mu_n^{\pi,p})_{n\in[N]}$, with $\mu_n^{\pi,p} \in \mathcal{S}_{\mathcal{X}\times\mathcal{A}}$. It can be calculated recursively for all $(x,a) \in \mathcal{X}$ and $n \in [N]$ by taking $\mu_0^{\pi,p}(x,a) = \mu_0(x,a)$ fixed, and

$$\mu_n^{\pi,p}(x,a) = \sum_{(x',a')\in\mathcal{X}\times\mathcal{A}} \mu_{n-1}^{\pi,p}(x',a')p_n(x|x',a')\pi_n(a|x). \tag{2}$$

**Offline CURL.** The classic CURL optimization problem in Eq. (1) considers minimizing a function $F : (\mathcal{S}_{\mathcal{X}\times\mathcal{A}})^N \to \mathbb{R}$, here defined as $F(\mu) := \sum_{n=1}^N f_n(\mu_n)$ with $f_n$ a convex function over $\mu_n$ with values in $[0,1]$, across all policies that induce $\mu^{\pi,p}$. Note that $F$ is not convex on the policy $\pi$. To convexify the problem, we define the set of state-action distributions satisfying the Bellman flow of a MDP with transition kernel $p$ as

$$\mathcal{M}_{\mu_0}^p := \left\{\mu \big| \sum_{a'\in\mathcal{A}} \mu_n(x',a') = \sum_{x\in\mathcal{X},a\in\mathcal{A}} p_n(x'|x,a)\mu_{n-1}(x,a) , \forall x' \in \mathcal{X}, \forall n \in [N]\right\}. \tag{3}$$

For any $\mu \in \mathcal{M}_{\mu_0}^p$, there exists a strategy $\pi$ such that $\mu^{\pi,p} = \mu$. It suffices to take $\pi_n(a|x) \propto \mu_n(x,a)$ when the normalization factor is non-zero, and arbitrarily defined otherwise. There is thus an equivalence between the CURL problem (optimization on policies) and a convex optimization problem on state-action distributions satisfying the Bellman flow:

$$\min_{\pi\in(\mathcal{S}_\mathcal{A})^{\mathcal{X}\times N}} F(\mu^{\pi,p}) \equiv \min_{\mu\in\mathcal{M}_{\mu_0}^p} F(\mu). \tag{4}$$

**Online CURL.** In this paper we consider the online CURL problem in a non-stationary setting. We assume a finite-horizon scenario with $T$ episodes. An oblivious adversary generates a sequence of changing objective functions $(F^t)_{t\in[T]}$, with $F^t$ being fully communicated to the learner only at the end of episode $t$. We assume $F^t$ is $L_F$-Lipschitz with respect to the $\|\cdot\|_{\infty,1}$ norm for all $t$. The

probability transition kernel is also allowed to evolve over time and is denoted by $p^t$ at episode $t$. The learner's objective is then to calculate a sequence of strategies $(\pi^t)_{t \in [T]}$ minimizing a total loss $L_T := \sum_{t=1}^{T} F^t(\mu^{\pi^t, p^t})$, while dealing with adversarial objective functions $F^t$ and changing probability transition kernels $p^t$. To measure the learner's performance, we use the notion of dynamic regret (the difference between the learner's total loss and that of any policy sequence $(\pi^{t,*})_{t \in [T]}$):

$$R_{[T]}\big((\pi^{t,*})_{t \in [T]}\big) := \sum_{t \in [T]} F^t(\mu^{\pi^t, p^t}) - F^t(\mu^{\pi^{t,*}, p^t}). \tag{5}$$

**Non-stationarity measures.** We consider the following two non-stationary measures $\Delta_\infty^p$ and $\Delta^p$ on the probability transition kernels that respectively measure abrupt and smooth variations:

$$\Delta_\infty^p := 1 + \sum_{t=1}^{T-1} \mathbb{1}_{\{p^t \neq p^{t+1}\}}, \quad \Delta^p := 1 + \sum_{t=1}^{T-1} \Delta_t^p, \quad \Delta_t^p := \max_{n,x,a} \|p_n^t(\cdot|x,a) - p_n^{t+1}(\cdot|x,a)\|_1. \tag{6}$$

Regarding dynamic regret, we define for any sequence of policies $(\pi^{t,*})_{t \in [T]}$, its non-stationarity measure as

$$\Delta^{\pi^*} := 1 + \sum_{t=1}^{T-1} \Delta_t^{\pi^*}, \qquad \Delta_t^{\pi^*} := \max_{n \in [N], x \in \mathcal{X}} \|\pi_n^{t,*}(\cdot|x) - \pi_n^{t+1,*}(\cdot|x)\|_1. \tag{7}$$

Moreover, for any interval $I \subseteq [T]$, we write $\Delta_I^p := \sum_{t \in I} \Delta_t^p$ and $\Delta_I^{\pi^*} := \sum_{t \in I} \Delta_t^{\pi^*}$.

**Dynamic's hypothesis.** For each episode $t$, let $(x_0^t, a_0^t) \sim \mu_0(\cdot)$, and for all time steps $n \in [N]$,

$$x_{n+1}^t = g_n(x_n^t, a_n^t, \epsilon_n^t), \tag{8}$$

where $g_n$ represents the deterministic part of the dynamics, and $(\epsilon_n^t)_{n \in [N]}$ is a sequence of independent external noises such that $\epsilon_n^t \sim h_n^t(\cdot)$, where $h_n^t$ is any centered distribution. Note that these dynamics imply that the probability transition kernel can be written as $p_{n+1}^t(x'|x,a) = \mathbb{P}\big(g_n(x,a,\epsilon_n^t) = x'\big)$. Different variants of this problem can be considered, depending on the prior information available about the dynamics in Eq. (8). In this article we consider the case where $g_n$ is fixed and known by the learner, but $h_n^t$ is unknown and can change (hence the source of uncertainty and non-stationarity of the transitions). To the best of our knowledge, there are no black-box algorithms in the literature that achieve sublinear regret for online CURL with adversarial losses without relying on model assumptions. In using RL methods to CURL, we believe model-optimistic approaches like UCRL (Upper Confidence RL [4]) could be adapted. However, these methods are computationally expensive, as they require solving an additional optimization problem in every episode. The black-box algorithm for CURL we consider from [39] provides closed-form solutions, which is more computationally efficient, but requires the same dynamic assumption as in Eq. (8). Another class of RL methods is policy optimization (PO), which directly optimizes the policy and often yields closed-form solutions, leading to faster performance. Recent theoretical work [37] has shown that PO methods can achieve near-optimal regret without model assumptions. However, these methods rely on RL's Bellman equations, which do not apply to CURL due to its non-linear nature. We believe that the MetaCURL analysis could potentially be extended to the case where $g_n$ is unknown but belongs to a parametric family. We leave this extension for future work.

This particular dynamic is also motivated by many real-world applications:

- Controlling a fleet of drones in a known environment, subject to external influences due to weather conditions or human interventions.
- Addressing data center power management aiming to cut energy expenses while maintaining service quality. Workload fluctuations cause dynamic job queue transitions, and volatile electricity prices lead to varying operational costs. The probabilities of task processing by each server are predetermined, but the probabilities of task arrival are uncertain [6].
- As renewable energy use increases and energy demand grows, balancing production and consumption becomes harder. Certain devices, like electric vehicle batteries and water heaters, can serve as flexible energy storage options. However, this requires electric grids to establish policies regulating when these devices turn on or off to match a desired consumption profile. These profiles can fluctuate daily due to changes in energy production levels. Despite knowing the devices' physical dynamics, household consumption habits remain unpredictable and constantly changing [51, 41].

**Outline.** In this paper, we propose a novel approach to handle non-stationarity in MDPs, being the first to propose a solution to CURL within this context. We begin in Section 3 by discussing the idea behind our algorithm's construction and the key challenges within our framework. Section 4 introduces MetaCURL, while Section 5 presents the main results of our regret analysis. The proofs' specifics are provided in the appendix.

## 3   Main idea

**A hypothetical learner who achieves optimal regret.** Let $m > 1$. Assume a hypothetical learner that could compute a sequence of restart times $1 = t_1 < \ldots < t_{m+1} = T + 1$, where for each $i \in [m]$ we let $\mathcal{I}_i := [t_i, t_{i+1} - 1]$, such that

$$\Delta^p_{\mathcal{I}_i} \leq \Delta^p/m. \tag{9}$$

Consider any algorithm that, when computing $(\pi^t)_{t \in I}$ with learning rate $\lambda$ for any interval $I \subseteq [T]$, attains a dynamic regret relative to any sequence of policies $(\pi^{t,*})_{t \in I}$ upper bounded by

$$R_I\big((\pi^{t,*})_{t \in I}\big) \leq c_1 \lambda |I| + \lambda^{-1}(c_2 \Delta^{\pi^*}_I + c_3) + |I| \Delta^p_I, \tag{10}$$

where $(c_j)_{j \in [3]}$ are constants that may depend on the MDP parameters, and on the interval size only in logarithmic terms. This kind of regret bound holds for Greedy MD-CURL from [39] as we show in Appendix G. Suppose the hypothetical learner could also access $\Delta^{\pi^*}$ to calculate the optimal learning rate. Hence, playing such an algorithm for all horizon $T$ with the optimal learning rate, the learner would have a dynamic regret upper bounded by

$$R_{[T]}\big((\pi^{t,*})_{t \in [T]}\big) \leq 2\sqrt{c_1(c_2 \Delta^{\pi^*} + c_3 m)T} + T \Delta^p m^{-1}.$$

Optimizing over $m$, the learner would obtain the optimal regret of order $\tilde{O}\big(\sqrt{\Delta^{\pi^*} T} + (\Delta^p)^{1/3} T^{2/3}\big)$. In the case where the MDP is piece-wise stationary, if the learner takes $\mathcal{I}_i$ such that $\Delta^p_{\mathcal{I}_i} = 0$, it obtains a regret of order $O(\sqrt{\Delta^{\pi^*} T} + \sqrt{\Delta^p_\infty T})$, where $\Delta^p_\infty$ is the number of times the probability transitions of the MDP change over $[T]$.

**A meta algorithm to learn restart times.** In reality, the restart times of Eq. (9), and the optimal learning rate, are unknown to the learner. Hence, we propose to build a meta aggregation algorithm to learn both. Let $\mathcal{E}$ represent a parametric black-box algorithm with dynamic regret as in Eq. (10). We introduce a meta algorithm $\mathcal{M}$ that, takes as input a finite set of learning rates $\Lambda$, and at each episode $t$, initializes $|\Lambda|$ instances of $\mathcal{E}$, denoted as $\mathcal{E}^{t,\lambda}$ for each $\lambda \in \Lambda$. Each $\mathcal{E}^{t,\lambda}$ operates independently within the interval $[t, T]$. At time $t$, $\mathcal{M}$ combines the decisions from the active runs $\{\mathcal{E}^{s,\lambda}\}_{s \leq t, \lambda \in \Lambda}$ by weighted average. The idea is that at time $t$, some of the outputs of $\{\mathcal{E}^{s,\lambda}\}_{s \leq t, \lambda \in \Lambda}$ are not based on data prior to $t' < t$, so if the environment changes at time $t'$, these outputs can be given a greater weight by the meta algorithm, enabling it to adapt more quickly to the change. At the same time, we expect a larger weight will be given to the empirically best learning rate. Let $\mathcal{M}(\mathcal{E}, \Lambda)$ be the complete algorithm.

**Remark 3.1.** *The meta-algorithm increases the computational complexity of the parametric black-box algorithm by a factor of $T \times |\Lambda|$, as it requires updating $t \times |\Lambda|$ instances at each episode $t$. By strategically designing intervals to run the black-box algorithms, previous works on online learning have reduced computational complexity to $O(\log(T))$ [15, 29, 27]. Extending our analysis to these intervals is straightforward, but it would complicate the presentation of the paper. Thus, we decided to present our results using the naive choice of intervals. Also, in Section 5, we show that a learning rate grid with $|\Lambda| = \log(T)$ is sufficient to obtain the optimal regret.*

**Regret decomposition.** Denote by $\pi^{t,s,\lambda}$ the policy output from $\mathcal{E}^{s,\lambda}$ at episode $t$, for learning rate $\lambda$, for all $s \leq t$, and by $\pi^t$ the policy output by the meta algorithm $\mathcal{M}(\mathcal{E}, \Lambda)$ to be played by the learner. The regret of $\mathcal{M}(\mathcal{E}, \Lambda)$ can be decomposed as the sum of the regret suffered by the meta algorithm aggregation scheme, $\mathcal{M}$, and the regret from the black-box algorithm, $\mathcal{E}$, played with any learning rate $\lambda \in \Lambda$. The dynamic regret, defined in Eq. (5), can be decomposed, for any set of intervals $\mathcal{I}_i = [t_i, t_{i+1} - 1]$, with $1 = t_1 < \ldots < t_{m+1} = T + 1$, and for any learning rate $\lambda \in \Lambda$, as

$$R_{[T]}\big((\pi^{t,*})_{t \in [T]}\big) = \underbrace{\sum_{i=1}^m \sum_{t \in \mathcal{I}_i} F^t(\mu^{\pi^t, p^t}) - F^t(\mu^{\pi^{t,t_i,\lambda}, p^t})}_{\text{Meta algorithm regret}} + \underbrace{\sum_{i=1}^m \sum_{t \in \mathcal{I}_i} F^t(\mu^{\pi^{t,t_i,\lambda}, p^t}) - F^t(\mu^{\pi^{t,*}, p^t})}_{\text{Black-box regret on } \mathcal{I}_i}$$

$$:= R^{\text{meta}}_{[T]} + R^{\text{black-box}}_{[T]}\big((\pi^{t,*})_{t \in [T]}\big). \tag{11}$$

The black-box regret on $\mathcal{I}_i$ is exactly the standard regret for $T = |\mathcal{I}_i|$ with a learning rate of $\lambda$. Hence, in order to prove low dynamic regret for $\mathcal{M}(\mathcal{E}, \Lambda)$ we have to: show that $\mathcal{M}$ incurs a low dynamic regret in each interval $\mathcal{I}_i$; find a black-box algorithm $\mathcal{E}$ for CURL that has dynamic regret as in Eq. (10), and build a learning rate grid $\Lambda$ allowing us to perform nearly as well as the optimal learning rate.

# 4 MetaCURL Algorithm

We call our meta-algorithm $\mathcal{M}$ MetaCURL. It is based on sleeping experts, is parameter-free, and achieves optimal regret. Its construction is described below.

## 4.1 Learning with expert advice

**General setting.** In Learning with Expert Advice (LEA), a learner makes sequential online predictions $u^1, \ldots, u^T$ in a decision space $\mathcal{U}$, over a series of $T$ episodes, with the help of $K$ experts [22, 36, 9]. For each round $t$, each expert $k$ makes a prediction $u^{t,k}$, and the learner combines the experts' predictions by computing a vector $v^t := (v^{t,1}, \ldots, v^{t,K}) \in \mathcal{S}_K$, and predicting the convex combination of experts' prediction $u^t := \sum_{k=1}^{K} v^{t,k} u^{t,k}$. The environment then reveals a convex loss function $\ell^t : \mathcal{U} \to \mathbb{R}$. Each expert suffers a loss $\ell^{t,k} := \ell^t(u^{t,k})$, and the learner suffers a loss $\hat{\ell}^t := \ell^t(u^t)$. The learner's objective is to keep the cumulative regret with respect to each expert as low as possible. For each expert $k$, this quantity is defined as $\text{Reg}_{[T]}(k) := \sum_{t=1}^{T} \hat{\ell}^t - \ell^{t,k}$.

**Sleeping experts.** In our case, each black-box algorithm is an expert that does not produce solutions outside its active interval. This problem can be reduced to the sleeping expert problem [8, 23], where experts are not required to provide solutions at every time step. Let $I^{t,k} \in \{0, 1\}$ define a signal equal to 1 if expert $k$ is active at episode $t$ and 0 otherwise. The algorithm knows $(I^{t,k})_{k \in [K]}$ and assigns a zero weight to sleeping experts ($I^{t,k} = 0$ implies $v^{t,k} = 0$). We would like to have a guarantee with respect to expert $k \in [K]$ but only when it is active. Hence, we now aim to bound a cumulative regret that depends on the signal $I^{t,k}$: $\text{Reg}_{[T]}^{\text{sleep}}(k) := \sum_{t=1}^{T} I^{t,k}(\hat{\ell}^t - \ell^{t,k})$. There is a generic reduction from the sleeping expert framework to the general LEA setting [3, 2] (see Appendix A.1).

## 4.2 Meta-aggregation scheme

In every episode $t$, for every learning rate $\lambda \in \Lambda$ and $s \leq t$, an instance $\mathcal{E}^{s,\lambda}$ of the black-box algorithm acts as an expert computing a policy $\pi^{t,s,\lambda}$. The meta algorithm aims to aggregate these predictions using a sleeping expert approach based on the expert's losses. However, within CURL's framework, the meta algorithm faces two challenges:

**Uncertainty.** At the episode's end, the learner has full information about the objective function $F^t$. If the learner also knew $p^t$, they could recursively calculate the corresponding state-action distribution $\mu^{\pi^{t,s,\lambda}, p^t}$ using Eq. (2) and observe the actual loss of each expert, denoted as $F^t(\mu^{\pi^{t,s,\lambda}, p^t})$. However, given that $p^t$ is unknown to the learner, the true loss remains unobservable. Consequently, the meta-algorithm needs to create an estimator $\hat{p}^t$ for $p^t$ and utilize it to estimate the losses. We propose a method to compute an estimator $\hat{p}^t$ in Subsection 4.3.

**Convexity.** As discussed in Section 2, the objective functions $F^t$ are not convex over the space of polices. However, CURL is equivalent to a convex problem over the state-action distributions satisfying the Bellman's flow as shown in Eq. (4). Therefore, instead of aggregating policies, the meta algorithm aggregates the associated state-action distributions using the probability estimator $\hat{p}^t$ and the recursive scheme at Eq. (2). We detail MetaCURL in Alg. 1 when employed with the Exponentially Weighted Average forecaster (EWA) as the sleeping expert subroutine (we detail EWA in Appendix A.2).

## 4.3 Building an estimator of $p^t$

As discussed earlier, applying the learning with experts framework requires estimating the loss of non-played expert policies, which depends on estimating the non-stationary transition probabilities $\hat{p}^t$. Standard RL techniques for bounding the $L_1$ norm between the empirical estimator $\hat{p}^t$ and the true

---

**Algorithm 1** MetaCURL with EWA

---

1: **Input:** number of episodes $T$, finite set of learning rates $\Lambda$, black-box algo. $\mathcal{E}$, EWA learning rate $\eta = \sqrt{8\log(T)T}$

2: **Initialization:** $\hat{p}_n^1(\cdot|x,a) := \frac{1}{|\mathcal{X}|}$ for all $n \in [N], (x,a) \in \mathcal{X} \times \mathcal{A}$

3: **for** $t = 1, \ldots, T$ **do**

4:  Start $|\Lambda|$ new instances of $\mathcal{E}$ denoted by $\mathcal{E}^{t,\lambda}$ for all $\lambda \in \Lambda$, assign each of them a new weight $v^{t,t,\lambda} = \frac{1}{|\Lambda|t}$, and normalize weight vectors $v^{t,s,\lambda}$ for $s \in [t-1]$ such that $v^t := (v^{t,s,\lambda})_{s \le t, \lambda \in \Lambda}$ is a probability vector in $\mathbb{R}^{t \times \Lambda}$

5:  For $s \le t$ and $\lambda \in \Lambda$, $\mathcal{E}^{s,\lambda}$ outputs $\pi^{t,s,\lambda}$

6:  Compute recursively $\mu^{\pi^{t,s,\lambda}, \hat{p}^t}$ using Eq. (2) for all $s \le t$ and $\lambda \in \Lambda$

7:  Aggregate the state-action distributions: $\mu^t := \sum_{s=1}^t \sum_{\lambda \in \Lambda} \mu^{\pi^{t,s,\lambda}, \hat{p}^t} v^{t,s,\lambda}$

8:  Retrieve $\pi^t$ from $\mu^t$: for all $n, (x,a)$,

$$\pi_n^t(a|x) = \begin{cases} \frac{\mu_n^t(x,a)}{\sum_{a' \in \mathcal{A}} \mu_n(x,a')}, & \text{if } \mu_n^t(x,a) \neq 0 \\ \frac{1}{|\mathcal{A}|}, & \text{if } \mu_n^t(x,a) = 0 \end{cases}$$

9:  Learner plays $\pi^t$: Agent starts at $(x_0^t, a_0^t) \sim \mu_0(\cdot)$

10:  **for** $n = 1, \ldots, N$ **do**

11:   Environment draws new state $x_n^t \sim p_n^t(\cdot|x_{n-1}^t, a_{n-1}^t)$

12:   Learner observes agent's external noise $\varepsilon_n^t$

13:   Agent chooses an action $a_n^t \sim \pi_n^t(\cdot|x_n^t)$

14:  **end for**

15:  Objective function $F^t$ is exposed

16:  Compute experts' losses $\ell^{t,s,\lambda} := F^t(\mu^{\pi^{t,s,\lambda}, \hat{p}^t})$, for all $s \le t$ and $\lambda \in \Lambda$

17:  Compute the new weight vector $v^{t+1}$: for all $s \le t$ and $\lambda \in \Lambda$,

$$v^{t+1,s,\lambda} = \frac{v^{t,s,\lambda} \exp\left(-\eta \ell^{t,s,\lambda}\right)}{\sum_{s'=1}^t \sum_{\lambda' \in \Lambda} v^{t,s',\lambda'} \exp\left(-\eta \ell^{t,s',\lambda'}\right)} \qquad \text{(EWA update)}$$

18:  Use agent's external noise trajectory $(\varepsilon_n^t)_{n \in [N]}$ to compute $\hat{p}^{t+1}$ as in Subsection 4.3

19: **end for**

---

dynamics $p^t$ [44, 49] are not applicable here due to non-stationarity. To address this, we introduce a second layer of sleeping experts for each $(n,x,a) \in [N] \times \mathcal{X} \times \mathcal{A}$, where each expert provides an empirical estimate of $p^t$ based on different intervals. We then propose a new loss function in Eq. (12) and conduct a novel regret analysis in Prop. 5.2 to achieve the optimal regret rate.

In each episode $t$, the learner calculates independent samples $x_{n,x,a}^t \sim p_n^t(\cdot|x,a)$ utilizing the external noise sequence $(\varepsilon_n^t)_{n \in [N]}$ observed (just let $x_{n,x,a}^t = g_{n-1}(x,a,\varepsilon_{n-1}^t)$, see Eq. (8)). Each expert outputs an empirical estimator of $p_n^t(\cdot|x,a)$ using samples across different intervals. We assume $T$ experts, with expert $s$ active in interval $[s,T]$. Expert $s$ at episode $t > s$ outputs:

$$\hat{p}_n^{t,s}(x'|x,a) = \frac{N_{n,x,a}^{s:t-1}(x')}{(t-s)}, \quad \text{with} \quad N_{n,x,a}^{s:t-1}(x') := \sum_{q=s}^{t-1} \mathbb{1}_{\{x_{n,x,a}^q = x'\}}.$$

We let $\hat{p}_n^t(\cdot|x,a)$ be the result of employing sleeping EWA with experts $\hat{p}_n^{t,s}(\cdot|x,a)$, for $s < t$. Typically, in density estimation with EWA, a logarithmic loss $-\log(\cdot)$ is used. However, in this case $-\log(\cdot)$ can be unbounded, so we opt here for a smoothed logarithmic loss, given by, for all $q \in \mathcal{S}_{\mathcal{X}}$,

$$\ell^t(q) := \sum_{x \in \mathcal{X}} -\log\left(q(x) + \frac{1}{|\mathcal{X}|}\right) \mathbb{1}_{\{\tilde{x}_{n,x,a}^t = x\}}, \quad \text{where } \tilde{x}_{n,x,a}^t \sim \left(p_n^t(\cdot|x,a) + \frac{1}{|\mathcal{X}|}\right)/2. \quad (12)$$

The definition of this non-standard loss is further clarified during the regret analysis in Section 5. This loss function is 1-exp concave (see Lemma 4 of [52]), hence the cumulative regret of EWA with respect to each expert $s \in [T]$, for all episodes $\tau \in [s,T]$, satisfies $\text{Reg}_{[s,\tau]}^{\text{sleep}}(s) = \sum_{t=s}^{\tau} \ell^t(\hat{p}_n^t(\cdot|x,a)) - \ell^t(\hat{p}_n^{t,s}(\cdot|,x,a)) \le \log(T)$ (for more information on the regret

bounds of EWA with exp-concave losses, see Appendix A.2). We describe the complete online scheme to compute $\hat{p}^t$ in Alg. 3 at Appendix B.

## 5 Regret analysis

This section presents the main result concerning MetaCURL's regret analysis. Subsection 5.1 shows an upper bound for $R^{\text{meta}}$ when MetaCURL is played with EWA and $\hat{p}^t$ is computed as in Subsection. 4.3. Subsection 5.2 introduces a learning rate grid for MetaCURL when the black-box algorithm meets the dynamic regret criteria in Eq. (10), providing an upper bound for $R^{\text{black-box}}$. Given the dynamic regret decomposition of Eq. (11), we see that the combination of these results leads to our main result, the full proof of which can be found in appendix (F) :

**Theorem 5.1** (Main result). *Let $\delta \in (0,1)$. Playing MetaCURL, with a parametric black-box algorithm $\mathcal{E}$ with dynamic regret as in Eq. (10), with a learning rate grid $\Lambda := \left\{ 2^{-j} | j = 0, 1, 2, \ldots, \lceil \log_2(T)/2 \rceil \right\}$, and with EWA as the sleeping expert subroutine, we obtain, with probability at least $1 - 2\delta$, for any sequence of policies $(\pi^{t,*})_{t\in[T]}$,*

$$R_{[T]}\big((\pi^{t,*})_{t\in[T]}\big) \leq \tilde{O}\Big(\sqrt{\Delta^{\pi^*}T} + \min\big\{\sqrt{T\Delta^p_\infty}, \ T^{2/3}(\Delta^p)^{1/3}\big\}\Big).$$

### 5.1 Meta-algorithm analysis

Given the uncertainty in the probability transition, the meta regret term can be decomposed as follows:

$$
\begin{aligned}
R_{[T]}^{\text{meta}} = &\underbrace{\sum_{t=1}^{T} F^t(\mu^{\pi^t, p^t}) - F^t(\mu^{\pi^t, \hat{p}^t})}_{R_{[T]}^{\hat{p}}(\pi^t)\ (\hat{p}^t\ \text{estimation})} \\
&+ \underbrace{\sum_{i=1}^{m} \sum_{t\in\mathcal{I}_i} F^t(\mu^{\pi^t, \hat{p}^t}) - F^t(\mu^{\pi^{t,t_i,\lambda}, \hat{p}^t})}_{\text{sleeping LEA regret}} + \underbrace{\sum_{i=1}^{m} \sum_{t\in\mathcal{I}_i} F^t(\mu^{\pi^{t,t_i,\lambda}, \hat{p}^t}) - F^t(\mu^{\pi^{t,t_i,\lambda}, p^t})}_{\sum_{i=1}^{m} \sum_{t\in\mathcal{I}_i} R_{\mathcal{I}_i}^{\hat{p}}(\pi^{t,t_i,\lambda})\ (\hat{p}^t\ \text{estimation})}.
\end{aligned}
\tag{13}
$$

**Sleeping LEA regret.** Referring to Thm. A.1 in Appendix A, using sleeping EWA as the sleeping expert subroutine of MetaCURL, with signals $I^{t,s} = 1$ for active experts ($s \leq t$), experts' convex losses $\ell^{t,s,\lambda} := F^t(\mu^{\pi^{t,s,\lambda}, \hat{p}^t})$, and learner loss $\hat{\ell}^t := F^t(\mu^{\pi^t, \hat{p}^t})$, yields, for any $m \in [T]$ and for any set of intervals $\mathcal{I}_i = [t_i, t_{i+1} - 1]$, with $1 = t_1 < \ldots < t_{m+1} = T + 1$,

$$
\begin{aligned}
\sum_{i=1}^{m} \sum_{t\in\mathcal{I}_i} F^t(\mu^{\pi^t, \hat{p}^t}) - F^t(\mu^{\pi^{t,t_i,\lambda}, \hat{p}^t}) &= \sum_{i=1}^{m} \text{Reg}_{\mathcal{I}_i}^{\text{sleep}}(t_i) \\
&\leq \sum_{i=1}^{m} \sqrt{\frac{|\mathcal{I}_i|}{2} \log(T|\Lambda|)} \leq \sqrt{\frac{mT}{2} \log(T|\Lambda|)}.
\end{aligned}
\tag{14}
$$

$\hat{p}^t$ **Estimation regret.** In a scenario without uncertainty in the MDP's probability transitions, the meta-algorithm's regret would simply be bounded by Eq. (14), the sleeping expert regret used as a subroutine. However, given the presence of uncertainty, the main challenge in analyzing the meta-regret comes from the regret terms associated with the estimator $\hat{p}^t$. We outline this analysis in Prop. 5.2.

**Proposition 5.2.** *Let $\delta \in (0,1)$, $C := \sqrt{\frac{1}{2} \log\big(\frac{N|\mathcal{X}||\mathcal{A}|2^{|\mathcal{X}|}T}{\delta}\big)}$, and $L_F$ be the Lipschitz constant of $F^t$, with respect to the norm $\|\cdot\|_{\infty,1}$, for all $t \in [T]$. With a probability of at least $1 - \delta$, MetaCURL obtains*

$$R_{[T]}^{\hat{p}}(\pi^t) := \sum_{t=1}^{T} F^t(\mu^{\pi^t, p^t}) - F^t(\mu^{\pi^t, \hat{p}^t}) \leq 2L_F N^2 |\mathcal{X}| \sqrt{3|\mathcal{A}|} C^{2/3} \log(T)^{1/3} T^{2/3} (\Delta^p)^{1/3}.$$

*For any $m \in [T]$ and for any set of intervals $\mathcal{I}_i = [t_i, t_{i+1} - 1]$, with $1 = t_1 < \ldots < t_{m+1} = T + 1$, the same bound is valid for $\sum_{i=1}^{m} \sum_{t\in\mathcal{I}_i} R_{\mathcal{I}_i}^{\hat{p}}(\pi^{t,t_i,\lambda})$.*

*Proof.* The proof idea is based mainly on the formulation of $\hat{p}^t$ described in Subsection 4.3. We start by using the convexity of $F^t$ to linearize the expression, then we apply Holder's inequality and exploit the $L_F$-Lipschitz property of $F^t$ to establish an upper bound based on the $L_1$ norm difference of the state-action distributions induced by the true probability transition and the estimator. Using Lemma C.5 in Appendix C, we then obtain that

$$R^{\hat{p}}_{[T]} \leq L_F \sum_{t=1}^{T} \sum_{n=1}^{N} \sum_{j=1}^{n} \sum_{x,a} \mu^{\pi^t,p^t}_{j-1}(x,a) \|p^t_j(\cdot|x,a) - \hat{p}^t_j(\cdot|x,a)\|_1.$$

To use the results from Subsection 4.3, we first regularize $p^t$ and $\hat{p}^t$, for each $(n,x,a)$, by averaging each with the uniform distribution over $\mathcal{X}$, that we denote by $p_0 := 1/|\mathcal{X}|$. As both probabilities are now lower bounded, we can employ Pinsker's inequality to convert the $L_1$ norm into a KL divergence. The sum over $t \in [T]$ of the KL divergence can then be decomposed as follows:

$$\sum_{t=1}^{T} \text{KL}\Big(\frac{p^t_j(\cdot|x,a) + p_0}{2} \Big| \frac{\hat{p}^t_j(\cdot|x,a) + p_0}{2}\Big) = \sum_{i=1}^{m} \sum_{t \in \mathcal{I}_i} \text{KL}\Big(\frac{p^t_j(\cdot|x,a) + p_0}{2} \Big| \frac{\hat{p}^{t,t_i}_j(\cdot|x,a) + p_0}{2}\Big)$$

$$+ \sum_{i=1}^{m} \sum_{t \in \mathcal{I}_i} \mathbb{E}_{\tilde{x}^t_{j,x,a}} \big[ \log\big(\hat{p}^{t,t_i}_j(\tilde{x}^t_{j,x,a}|x,a) + p_0\big) - \log\big(\hat{p}^t_j(\tilde{x}^t_{j,x,a}|x,a) + p_0\big)\big],$$

where $\hat{p}^{t,t_i}_j(\cdot|x,a)$ is the empirical estimate of $p^t_j(\cdot|x,a)$ calculated with the observed data from $t_i$ to $t-1$, and the expectation is over $\tilde{x}^t_{j,x,a} \sim (p^t_j(\cdot|x,a) + p_0)/2$. The second term is the cumulative regret of computing $\hat{p}^t$ using EWA with loss as in Eq. (12), and is bounded by $m \log(T)$. We finish and give more details of the proof in Appendix D. $\qquad\square$

Prop. 5.2 together with Eq. (14) yields the main result of this subsection:

**Proposition 5.3** (Meta regret bound). *With the same assumptions as Prop. 5.2, for any $m \in [T]$, with probability at least $1 - 2\delta$,*

$$R^{meta}_{[T]} \leq 4L_F N^2 |\mathcal{X}| \sqrt{3|\mathcal{A}|} C^{2/3} \log(T)^{1/3} T^{2/3} (\Delta^p)^{1/3} + \sqrt{\frac{mT}{2} \log(T|\Lambda|)}.$$

### 5.2 Black-box algorithm analysis

Assuming $\mathcal{E}$ is a parametric black-box algorithm with dynamic regret satisfying Eq. (10) for any learning rate $\lambda > 0$, we only need to address the selection of the $\lambda$s grid and optimize across $\lambda$ to achieve our final bound on $R^{\text{black-box}}_{[T]}$.

**Learning rate grid.** The dynamic regret of Eq. (10) implies that any two $\lambda$ that are a constant factor of each other will guarantee the same upper-bound up to essentially the same constant factor. We therefore choose an exponentially spaced grid

$$\Lambda := \big\{2^{-j} | j = 0, 1, 2, \ldots, \lceil \log_2(T)/2 \rceil\big\}. \tag{15}$$

The meta-algorithm aggregation scheme guarantees that the learner performs as well as the best empirical learning rate. We thus obtain a bound on $R^{\text{black-box}}_{[T]}$, with its proof in Appendix E:

**Proposition 5.4** (Black-box regret bound). *Assume MetaCURL is played with a black-box algorithm satisfying dynamic regret as in Eq. (10), with learning rate grid as in Eq. (15). Hence, for any sequence of policies $(\pi^{t,*})_{t \in [T]}$,*

$$R^{black\text{-}box}_{[T]}\big((\pi^{t,*})_{t \in [T]}\big) \leq N\Big(\frac{c_2 \Delta^{\pi^*} + c_3}{c_1}\Big) + c_1 \sqrt{T} + 3\sqrt{c_1(c_2 \Delta^{\pi^*} + c_3 m)T} + \frac{T\Delta^p}{m}.$$

**Greedy MD-CURL.** Greedy MD-CURL, developed by [39], is a computationally efficient policy-optimization algorithm known for achieving sublinear static regret in online CURL with adversarial objective functions within a stationary MDP. In Thm. G.3 of Appendix G, we extend this analysis showing that Greedy MD-CURL also achieves dynamic regret as in Eq. (10). To our knowledge, this is the first dynamic regret result for a CURL algorithm. Hence, Greedy MD-CURL can be used as a black-box for MetaCURL. We detail Greedy MD-CURL in Alg. 4 in Appendix G.

# 6 Conclusion, discussion, and future work

In this paper, we present MetaCURL, the first algorithm for dealing with non-stationarity in CURL, a setting covering many problems in the literature that modifies the standard linear RL configuration, making typical RL techniques difficult to use. We also employ a novel approach to deal with non-stationarity in MDPs using the learning with expert advice framework from the online learning literature. The main difficulty in analyzing this method arises from uncertainty about probability transitions. We overcome this problem by employing a second expert scheme, and show that MetaCURL achieves near-optimal regret.

Compared to the RL literature, our approach is more efficient, deals with adversarial losses, and has a better regret dependency concerning the varying losses, but to do so, we need to simplify the assumptions about the dynamics (all uncertainty comes only from the external noise, that is independent of the agent's state-action). There seems to be a trade-off in RL: all algorithms dealing with both non-stationarity and full exploration use UCRL-type approaches, and are thus computationally expensive. We thus leave a question for future work: *How can we effectively manage non-stationarity and adversarial losses using efficient algorithms, all while addressing full exploration?*

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

# A Learning with expert advice

In this section, we take a closer look at the Learning with Expert Advice (LEA) framework. We start by presenting, in Subsection A.1, a general reduction of the sleeping experts framework to the standard framework. Thus, any LEA algorithm can be used as a sub-routine for MetaCURL. In Section 5 of the main paper, we show a regret bound for MetaCURL using the Exponentially Weighted Average Forecaster (EWA) algorithm [9], also known as Hedge. In Subsection A.2 we present the main results of playing EWA with convex and exp-concave losses.

**Setting.** We recall the general setting of learning with expert advice (LEA) as presented in the main paper: a learner makes sequential online predictions $u^1, \ldots, u^T$ in a decision space $\mathcal{U}$, over a series of $T$ episodes, with the help of $K$ experts. For each round $t$, each expert $k$ makes a prediction $u^{t,k}$, and the learner combines the experts' predictions by computing a vector $v^t := (v^{t,1}, \ldots, v^{t,K}) \in \mathcal{S}_K$, and predicting $u^t := \sum_{k=1}^K v^{t,k} u^{t,k}$. The environment then reveals a convex loss function $\ell^t : \mathcal{U} \to \mathbb{R}$. Each expert suffers a loss $\ell^{t,k} := \ell^t(u^{t,k})$, and the learner suffers a loss $\hat{\ell}^t := \ell^t(u^t)$. The learner's objective is to keep the cumulative regret with respect to each expert as low as possible. For each expert $k$, this quantity is defined as $\text{Reg}_{[T]}(k) := \sum_{t=1}^T \hat{\ell}^t - \ell^{t,k}$.

## A.1 Sleeping experts

The sleeping expert problem [8, 23] is a LEA framework where experts are not required to provide solutions at every time step. Let $I^{t,k} \in \{0,1\}$ define a binary signal that equals 1 if expert $k$ is active at episode $t$ and 0 otherwise. The algorithm knows $(I^{t,k})_{k \in [K]}$ and assigns a zero weight to sleeping experts. We would like to have a guarantee with respect to expert $k \in [K]$ but only when it is active. Hence, we now aim to bound a cumulative regret that depends on the signal $I^{t,k}$: $\text{Reg}_{[T]}^{\text{sleep}}(k) := \sum_{t=1}^T I^{t,k}(\hat{\ell}^t - \ell^{t,k})$. We present a generic reduction from the sleeping expert framework to the standard LEA framework [3, 2]:

Let, for all episodes $t \in [T]$,

$$\hat{u}^t := \frac{\sum_{k=1}^K I^{t,k} v^{t,k} u^{t,k}}{\sum_{k=1}^K I^{t,k} v^{t,k}}.$$

We play a standard LEA algorithm with modified outputs where, at episode $t$, expert $k$ outputs

$$\tilde{u}^{t,k} := \begin{cases} u^{t,k}, & \text{if } k \text{ is active at episode } t \\ \hat{u}^t, & \text{if not.} \end{cases}$$

A standard LEA algorithm gives an upper bound on the regret $\text{Reg}_T(k)$ with respect to each expert $k$. Using that $\sum_{k=1}^K \tilde{u}^{t,k} v^{t,k} = \hat{u}^t$, we obtain that

$$
\begin{aligned}
\text{Reg}_{[T]}(k) &:= \sum_{t=1}^T \ell^t \left( \sum_{k=1}^K \tilde{u}^{t,k} v^{t,k} \right) - \ell^t(\tilde{u}^{t,k}) \\
&= \sum_{t=1}^T \ell^t(\hat{u}^t) - \ell^t(\tilde{u}^{t,k}) \\
&= \sum_{t=1}^T I^{t,k} \left( \ell^t(\hat{u}^t) - \ell^t(u^{t,k}) \right) \\
&=: \text{Reg}_{[T]}^{\text{sleep}}(k).
\end{aligned}
$$

Consequently, the cumulative regret with respect to each expert during the times it is active is upper bounded by the standard regret of playing a LEA algorithm with the modified outputs.

## A.2 Exponentially Weighted Average forecaster

The exponentially weighted average forecaster (EWA), also called Hedge, is a LEA algorithm that chooses a weight that decreases exponentially fast with past errors. We present EWA in Alg. 2.

---
**Algorithm 2** EWA (Exponentially Weighted Average)
---
**Input:** $[K] := \{1, \ldots, K\}$ a finite set of experts, $v^0$ a prior over $[K]$, a learning rate $\eta > 0$
**for** $t \in \{1, \ldots, T\}$ **do**
    Observe loss function $\ell^t$, compute the loss suffered by each expert $\ell^{t,k} := \ell^t(u^{t,k})$ and suffer loss $\hat{\ell}^t := \ell\left( \sum_{k=1}^{K} v^{t,k} u^{t,k} \right)$
    Update for all $k \in [K]$ :

$$v^{t+1,k} = \frac{v^{t,k} \exp\left(-\eta \ell^{t,k}\right)}{\sum_{k'=1}^{K} v^{t,k'} \exp\left(-\eta \ell^{t,k'}\right)}$$

**end for**
---

We recall two results of playing EWA with convex losses, and with exp-concave losses, used in the main paper:

**Theorem A.1** (EWA with convex losses: Corollary 2.2 from [9]). *If the $\ell^t$ losses are convex and take value in $[0, 1]$, then the regret of the learner playing EWA with any $\eta > 0$ satisfies, for any $k \in [K]$,*

$$Reg_{[T]}(k) \leq \frac{\log(K)}{\eta} + \frac{T\eta}{8}.$$

*In particular, with $\eta = \sqrt{8\log(K)/T}$, the upper bound becomes $\sqrt{(T/2)\log(K)}$.*

**Theorem A.2** (EWA with exp-concave losses: Thm. 3.2 from [9]). *If the $\ell^t$ losses are $\eta$-exp concave, then the regret of the learner playing EWA (with the same value of $\eta$) satisfies, for any $k \in [K]$,*

$$Reg_{[T]}(k) \leq \frac{\log(K)}{\eta}.$$

# B   Algorithm for the online estimation of the probability kernel ($\hat{p}^t$ estimator)

---

**Algorithm 3** Online estimation of the probability kernel ($\hat{p}^t$ estimator)

---

1: **for** $t \in \{1, \ldots, T\}$ **do**
2:    Get agent's external noise trajectory $(\varepsilon_n^t)_{n \in [N]}$ from MetaCURL
3:    **for** $(n, x, a) \in [N] \times \mathcal{X} \times \mathcal{A}$ **do**
4:       Compute $x_{n,x,a}^t = g_{n-1}(x, a, \varepsilon_{n-1}^t)$
5:       Update the empirical estimations for $s < t$ and $x' \in \mathcal{X}$:

$$\hat{p}_n^{t,s}(x'|x,a) = \frac{\mathbb{1}_{\{x_{n,x,a}^t = x'\}}}{t-s} + \frac{(t-1-s)}{t-s} \hat{p}_n^{t-1,s}(x'|x,a)$$

6:       Initialize a new estimator, $\hat{p}_n^{t,t}(x'|x,a) = \mathbb{1}_{\{x_{n,x,a}^t = x'\}}$ for all $x' \in \mathcal{X}$, assign it a new weight vector $v_{n,x,a}^{t,t} = \frac{1}{t}$, and normalize weight vectors $v_{n,x,a}^{t,s}$ for $s \in [t-1]$ such that $v_{n,x,a}^t := (v_{n,x,a}^{t,s})_{s \leq t, \lambda \in \Lambda}$ is a probability vector in $\mathbb{R}^t$
7:       Simulate a sample $\tilde{x}_{n,x,a}^t$ from distribution $\left(p_n^t(\cdot|x,a) + \frac{1}{|\mathcal{X}|}\right)/2$:

$$\tilde{x}_{n,x,a}^t = \begin{cases} x_{n,x,a}^t, & \text{with probability } 1/2, \\ \text{Uniformly over } \mathcal{X}, & \text{with probability } 1/2, \end{cases}$$

and use it to build the loss function

$$\ell^t(q) := \sum_{x \in \mathcal{X}} -\log\left(q(x) + \frac{1}{|\mathcal{X}|}\right)\mathbb{1}_{\{\tilde{x}_{n,x,a}^t = x\}}$$

8:       Update weights using EWA with loss $\ell^t$: for all $s \leq t$,

$$v_{n,x,a}^{t+1,s} = \frac{\hat{v}_{n,x,a}^{t,s} \exp\left(-\ell^t(\hat{p}_n^{t,s}(\cdot|x,a))\right)}{\sum_{s'=1}^t \hat{v}_{n,x,a}^{t,s'} \exp\left(-\ell^t(\hat{p}_n^{t,s'}(\cdot|x,a))\right)} \qquad \text{(EWA update)}$$

9:       Compute $\hat{p}_n^{t+1}(\cdot|x,a) = \sum_{s=1}^t v_{n,x,a}^{t+1,s} \hat{p}_n^{t,s}(\cdot|x,a)$
10:   **end for**
11:   Issue $\hat{p}^{t+1}$ to MetaCURL (line 18 of Alg. 1)
12: **end for**

---

# C   Auxiliary lemmas

We start with some auxiliary results. For $t \in I := [t_s + 1, t_e] \subseteq [T]$, we define the average probability distribution for all $n$ and $(x, a)$ as

$$\overline{p}^t(x'|x,a) = \frac{1}{t - t_s} \sum_{s=t_s}^{t-1} p_n^s(x'|x,a).$$

**Lemma C.1.** *Let $\hat{p}^{t,t_s}$ be the empirical probability transition kernel computed with data from episodes $[t_s, t-1]$. For any $\delta \in (0,1)$, with probability $1 - \delta$,*

$$\|\hat{p}_n^{t,t_s}(\cdot|x,a) - \overline{p}_n^t(\cdot|x,a)\|_1 \leq \sqrt{\frac{1}{2(t - t_s)} \log\left(\frac{N|\mathcal{X}||\mathcal{A}|2^{|\mathcal{X}|}T}{\delta}\right)},$$

*simultaneously for all $n \in [N]$, $(x, a) \in \mathcal{X} \times \mathcal{A}$, $t_s \in [T-1]$, and $t \in [t_s + 1, T]$.*

*Proof.* For a fixed $n \in [N]$, $(x, a) \in \mathcal{X} \times \mathcal{A}$, and $\theta \in \{-1, 1\}^{|\mathcal{X}|}$, we define for all $s \in I$,

$$Y_{n,x,a,\theta}^s := \sum_{x' \in \mathcal{X}} \theta(x')\mathbb{1}_{\{g_n(x,a,\varepsilon_n^s)=x'\}},$$

a Bernoulli random variable with mean value given by $\sum_{x' \in \mathcal{X}} \theta(x')p_n^s(x'|x,a)$.

The sequence of random variables given by $\left(Y_{n,x,a,\theta}^s\right)_{s\in I}$ is independent, as we assume that the external noises observed at each episode are all independent. Hence, by Hoeffding's inequality we get that, for all $\xi > 0$,

$$\mathbb{P}\left(\sum_{s=t_s}^{t-1} Y_{n,x,a,\theta}^s - \mathbb{E}\left[\sum_{s=t_s}^{t-1} Y_{n,x,a,\theta}^s\right] \geq \xi\right) \leq \exp\left(\frac{-2\xi^2}{t-t_s}\right).$$

Therefore, we have that

$$\mathbb{P}\left(\sum_{x'\in\mathcal{X}} \theta(x')\left(\hat{p}_n^{t,t_s}(x'|x,a) - \bar{p}_n^t(x'|x,a)\right) \geq \xi\right)$$

$$= \mathbb{P}\left(\frac{1}{t-t_s}\left[\sum_{s=t_s}^{t-1}\sum_{x'\in\mathcal{X}} \theta(x')\mathbb{1}_{\{g_n(x,a,\varepsilon_n^s)=x'\}} - \sum_{s=t_s}^{t-1}\theta(x')p_n^s(x'|x,a)\right] \geq \xi\right)$$

$$= \mathbb{P}\left(\frac{1}{t-t_s}\left(\sum_{s=t_s}^{t-1} Y_{n,x,a,\theta}^s - \mathbb{E}\left[\sum_{s=t_s}^{t-1} Y_{n,x,a,\theta}^s\right]\right) \geq \xi\right)$$

$$\leq \exp\left(-2\xi^2(t-t_s)\right).$$

By applying an union bound on all $n \in [N]$, $(x,a) \in \mathcal{X} \times \mathcal{A}$, and $\theta \in \{-1,1\}^{|\mathcal{X}|}$ and noting that, for any two probability distributions $p,q \in \Delta_{\mathcal{X}}$, we have that

$$\|p-q\|_1 = \max_{\theta\in\{-1,1\}^{|\mathcal{X}|}} \sum_{x\in\mathcal{X}} \theta(x)(p(x)-q(x)),$$

we arrive at the final result.

$\square$

**Lemma C.2.** *Let $t \in I := [t_s + 1, t_e] \subseteq [T]$. For all $n \in [N]$, and $(x,a) \in \mathcal{X} \times \mathcal{A}$,*

$$\|p_n^t(\cdot|x,a) - \bar{p}_n^t(\cdot|x,a)\|_1 \leq \sum_{j=t_s}^{t-1} \Delta_j^p,$$

*Proof.* For $t \in I$, and for all $n$ and $(x,a)$ we have that

$$\|p_n^t(\cdot|x,a) - \bar{p}_n^t(\cdot|x,a)\|_1 = \sum_{x'\in\mathcal{X}} \left|p_n^t(x'|x,a) - \frac{1}{t-t_s}\sum_{s=t_s}^{t-1} p_n^s(x'|x,a)\right|$$

$$= \sum_{x'\in\mathcal{X}} \frac{1}{t-t_s}\left|\sum_{s=t_s}^{t-1}\left(p_n^t(x'|x,a) - p_n^s(x'|x,a)\right)\right|$$

$$\leq \frac{1}{t-t_s}\sum_{s=t_s}^{t-1}\sum_{j=s+1}^{t} \|p_n^j(\cdot|x,a) - p_n^{j-1}(\cdot|x,a)\|_1$$

$$= \frac{1}{t-t_s}\sum_{j=t_s}^{t-1}(j-t_s)\|p_n^j(\cdot|x,a) - p^{j-1}(\cdot|x,a)\|_1$$

$$\leq \sum_{j=t_s}^{t-1} \Delta_j^p,$$

where recall that we define $\Delta_j^p := \max_{n,s,a} \|p_n^{j+1}(\cdot|x,a) - p_n^j(\cdot|x,a)\|_1$. $\square$

**Lemma C.3.** *Let $\hat{p}^{t,t_s}$ be the empirical probability transition kernel computed with data from episodes $[t_s, t-1]$. For any $\delta \in (0,1)$, with probability $1-\delta$,*

$$\|p_n^t(\cdot|x,a) - \hat{p}_n^{t,t_s}(\cdot|x,a)\|_1 \leq \sqrt{\frac{1}{2(t-t_s)}\log\left(\frac{N|\mathcal{X}||\mathcal{A}|2^{|\mathcal{X}|}T}{\delta}\right)} + \sum_{j=t_s}^{t-1}\Delta_j^p,$$

*simultaneously for all $n \in [N]$, $(x,a) \in \mathcal{X} \times \mathcal{A}$, $t_s \in [T-1]$, and $t \in [t_s + 1, T]$.*

*Proof.* The result follows immediately from the triangular inequality and Lemmas C.1 and C.2. $\quad\square$

**Lemma C.4** (A version of the inverse of Pinsker's inequality)**.** *Let $p', q'$ be any distributions over $\mathcal{S}_{\mathcal{X}}$. Define*

$$p := \frac{p' + \frac{1}{|\mathcal{X}|}}{2}, \quad and \quad q := \frac{q' + \frac{1}{|\mathcal{X}|}}{2}.$$

*Therefore,*

$$KL(p \mid q) \leq 2|\mathcal{X}|\|p - q\|_1^2.$$

*Proof.* First, note that $q$ is lower bounded by $\frac{1}{2|\mathcal{X}|}$, hence $\mathrm{KL}(p \mid q)$ is well defined. Also, by convexity of the simplex, $p, q \in \mathcal{S}_{\mathcal{X}}$, therefore

$$
\begin{aligned}
\mathrm{KL}(p \mid q) &= \sum_{x \in \mathcal{X}} p(x) \log\left(\frac{p(x)}{q(x)}\right) \\
&= \sum_{x \in \mathcal{X}} p(x) \log\left(1 + \left(\frac{p(x)}{q(x)} - 1\right)\right) \\
&\leq \sum_{x \in \mathcal{X}} p(x) \left(\frac{p(x)}{q(x)} - 1\right) \\
&= \sum_{x \in \mathcal{X}} \frac{p(x)}{q(x)}\big(p(x) - q(x)\big) + \sum_{x \in \mathcal{X}} \big(q(x) - p(x)\big) \\
&= \sum_{x \in \mathcal{X}} \frac{\big(p(x) - q(x)\big)^2}{q(x)} \\
&\leq \frac{1}{\min_{x \in \mathcal{X}} q(x)} \|p - q\|_1^2 \\
&\leq 2|\mathcal{X}|\|p - q\|_1^2.
\end{aligned}
$$

$\square$

**Lemma C.5.** *For any strategy $\pi \in (\mathcal{S}_{\mathcal{A}})^{\mathcal{X} \times N}$, for any two probability kernels $p = (p_n)_{n \in [N]}$ and $q = (q_n)_{n \in [N]}$ such that $p_n, q_n : \mathcal{X} \times \mathcal{A} \times \mathcal{X} \rightarrow [0, 1]$, and for all $n \in [N]$,*

$$\|\mu_n^{\pi,p} - \mu_n^{\pi,q}\|_1 \leq \sum_{i=0}^{n-1} \sum_{x,a} \mu_i^{\pi,p}(x, a) \|p_{i+1}(\cdot|x, a) - q_{i+1}(\cdot|x, a)\|_1.$$

*Proof.* From the definition of a state-action distribution sequence induced by a policy $\pi$ in a MDP with probability kernel $p$ in Eq. (2), we have that for all $(x, a) \in \mathcal{X} \times \mathcal{A}$ and $n \in [N]$,

$$\mu_n^{\pi,p}(x, a) = \sum_{x',a'} \mu_{n-1}^{\pi,p}(x', a') p_n(x|x', a') \pi_n(a|x).$$

Thus,

$$\|\mu_n^{\pi,p} - \mu_n^{\pi,q}\|_1 = \sum_{x,a} |\mu_n^{\pi,p}(x,a) - \mu_n^{\pi,q}(x,a)|$$

$$= \sum_{x,a} \sum_{x',a'} |\mu_{n-1}^{\pi,p}(x',a')p_n(x|x',a') - \mu_{n-1}^{\pi,q}(x',a')q_n(x|x',a')|\pi_n(a|x)$$

$$= \sum_{x} \sum_{x',a'} |\mu_{n-1}^{\pi,p}(x',a')p_n(x|x',a') - \mu_{n-1}^{\pi,q}(x',a')q_n(x|x',a')|$$

$$= \sum_{x} \sum_{x',a'} |\mu_{n-1}^{\pi,p}(x',a')p_n(x|x',a') - \mu_{n-1}^{\pi,p}(x',a')q_n(x|x',a')$$

$$+ \mu_{n-1}^{\pi,p}(x',a')q_n(x|x',a') - \mu_{n-1}^{\pi,q}(x',a')q_n(x|x',a')|$$

$$\leq \sum_{x',a'} \mu_{n-1}^{\pi,p}(x',a')\|p_n(\cdot|x',a') - q_n(\cdot|x',a')\|_1 + \sum_{x',a'} |\mu_{n-1}^{\pi,p}(x',a') - \mu_{n-1}^{\pi,q}(x',a')|$$

$$= \sum_{x',a'} \mu_{n-1}^{\pi,p}(x',a')\|p_n(\cdot|x',a') - q_n(\cdot|x',a')\|_1 + \|\mu_{n-1}^{\pi,p} - \mu_{n-1}^{\pi,q}\|_1.$$

Since for $n = 0$, $\|\mu_0^{\pi,p} - \mu_0^{\pi,q}\|_1 = 0$, by induction we get that

$$\|\mu_n^{\pi,p} - \mu_n^{\pi,q}\|_1 \leq \sum_{i=0}^{n-1} \sum_{x',a'} \mu_i^{\pi,p}(x',a')\|p_{i+1}(\cdot|x',a') - q_{i+1}(\cdot|x',a')\|_1.$$

$\square$

**Lemma C.6.** *For any pair of strategies $\pi, \pi' \in (\Delta_A)^{\mathcal{X} \times N}$, for any probability kernel $p = (p_n)_{n \in [N]}$ such that $p_n : \mathcal{X} \times \mathcal{A} \times \mathcal{X} \to [0,1]$, and for all $n \in [N]$,*

$$\|\mu_n^{\pi,p} - \mu_n^{\pi',p}\|_1 \leq \sum_{i=1}^{n} \sum_{x \in \mathcal{X}} \rho_i^{\pi,p}(x)\|\pi_i(\cdot|x) - \pi_i'(\cdot|x)\|_1,$$

*where $\rho_i^{\pi,p}(x) := \sum_{a \in \mathcal{A}} \mu_i^{\pi,p}(x,a)$ for all $x \in \mathcal{X}$ and $i \in [N]$.*

*Proof.* Using the recursive relation from Eq. (2) of a state-action distribution induced by a policy $\pi$ in a MDP with probability transition $p$ we have that

$$\|\mu_n^{\pi,p} - \mu_n^{\pi',p}\|_1 = \sum_{x,a} |\mu_n^{\pi,p}(x,a) - \mu_n^{\pi',p}(x,a)|$$

$$\leq \sum_{x,a} \sum_{x',a'} |\mu_{n-1}^{\pi,p}(x',a')\pi_n(a|x) - \mu_{n-1}^{\pi',p}(x',a')\pi_n'(a|x)|p_n(x|x',a')$$

$$\leq \sum_{x,a} \sum_{x',a'} \mu_{n-1}^{\pi,p}(x',a')p_n(x|x',a')|\pi_n(a|x) - \pi_n'(a|x)|$$

$$+ \sum_{x,a} \sum_{x',a'} \pi_n'(a|x)p_n(x|x',a')|\mu_{n-1}^{\pi,p}(x',a') - \mu_{n-1}^{\pi',p}(x',a')|$$

$$= \sum_{x} \rho_n^{\pi,p}(x)\|\pi_n(\cdot|x) - \pi_n'(\cdot|x)\|_1 + \|\mu_{n-1}^{\pi,p} - \mu_{n-1}^{\pi',p}\|_1.$$

Since $\mu_0$ is fixed for each state-action distribution sequence, by induction we obtain that

$$\|\mu_n^{\pi,p} - \mu_n^{\pi',p}\|_1 \leq \sum_{i=1}^{n} \sum_{x} \rho_i^{\pi,q}(x)\|\pi_i^t(\cdot|x) - \pi_i^{t-1}(\cdot|x)\|_1,$$

completing the proof. $\square$

# D  Proof of Prop. 5.2: $R^{\hat{p}}_{[T]}$ regret analysis

*Proof.* Here, we set an upper bound on the term $R^{\hat{p}}_{[T]}$ where we pay for errors in estimating $p^t$ by $\hat{p}^t$.

$$
R^{\hat{p}}_{[T]} := \sum_{t=1}^{T} F^t(\mu^{\pi^t, p^t}) - F^t(\mu^{\pi^t, \hat{p}^t}) \leq \sum_{t=1}^{T} \langle \nabla F^t(\mu^{\pi^t, p^t}), \mu^{\pi^t, p^t} - \mu^{\pi^t, \hat{p}^t} \rangle
$$

$$
\leq L_F \sum_{t=1}^{T} \sum_{n=1}^{N} \| \mu_n^{\pi^t, p^t} - \mu_n^{\pi^t, \hat{p}^t} \|_1
$$

$$
\leq L_F \sum_{t=1}^{T} \sum_{n=1}^{N} \sum_{j=1}^{n} \sum_{x,a} \mu_{j-1}^{\pi^t, p^t}(x,a) \| p_j^t(\cdot|x,a) - \hat{p}_j^t(\cdot|x,a) \|_1 .
$$

To obtain the first inequality, we use the convexity of $F^t$ for all $t \in [T]$, then we use Holder's inequality and the fact that $F^t$ is $L_F$-Lipschitz, and for the last inequality we use Lemma C.5.

The difficulty in analyzing the $L_1$ difference between $p^t$ and $\hat{p}^t$ arises from the non-stationarity of $p^t$. To overcome this we want to use the scheme presented in Subsection 4.3. By Cauchy-Schwartz, we get that

$$
R^{\hat{p}}_{[T]} \leq L_F \sqrt{ \underbrace{\sum_{t=1}^{T} \sum_{n=1}^{N} \sum_{j=1}^{n} \sum_{x,a} (\mu_{j-1}^{\pi^t, p^t}(x,a))^2}_{\leq TN^2} \underbrace{\sum_{t=1}^{T} \sum_{n=1}^{N} \sum_{j=1}^{n} \sum_{x,a} \| p_j^t(\cdot|x,a) - \hat{p}_j^t(\cdot|x,a) \|_1^2}_{(*)} } . \tag{16}
$$

**Analysis of the $L_1$ norm.** We start by analysing the sum over $t \in [T]$ of the $L_1$ norm in term $(*)$. For each $n \in [N], j \in [n]$ and $(x,a) \in \mathcal{X} \times \mathcal{A}$,

$$
\sum_{t=1}^{T} \| p_j^t(\cdot|x,a) - \hat{p}_j^t(\cdot|x,a) \|_1^2 = 4 \sum_{t=1}^{T} \left\| \frac{p_j^t(\cdot|x,a) + \frac{1}{|\mathcal{X}|}}{2} - \left( \frac{\hat{p}_j^t(\cdot|x,a) + \frac{1}{|\mathcal{X}|}}{2} \right) \right\|_1^2
$$

$$
\leq 8 \sum_{t=1}^{T} \mathrm{KL} \left( \frac{p_j^t(\cdot|x,a) + \frac{1}{|\mathcal{X}|}}{2} \bigg| \frac{\hat{p}_j^t(\cdot|x,a) + \frac{1}{|\mathcal{X}|}}{2} \right),
$$

where we apply Pinsker's inequality.

Consider a sequence of episodes $1 = t_1 < t_2 < \ldots < t_{m+1} = T+1$, with $\mathcal{I}_i := [t_i, t_{i+1} - 1]$, such that $\Delta^p_{\mathcal{I}_i} \leq \Delta^p / m$ for all $i \in [m]$. Decomposing the KL sum over $t \in [T]$ as a sum on the intervals $\mathcal{I}_i$, we obtain that

$$
\sum_{t=1}^{T} \mathrm{KL} \left( \frac{p_j^t(\cdot|x,a) + \frac{1}{|\mathcal{X}|}}{2} \bigg| \frac{\hat{p}_j^t(\cdot|x,a) + \frac{1}{|\mathcal{X}|}}{2} \right) = \underbrace{\sum_{i=1}^{m} \sum_{t \in \mathcal{I}_i} \mathrm{KL} \left( \frac{p_j^t(\cdot|x,a) + \frac{1}{|\mathcal{X}|}}{2} \bigg| \frac{\hat{p}_j^{t,t_i}(\cdot|x,a) + \frac{1}{|\mathcal{X}|}}{2} \right)}_{(i)}
$$

$$
+ \underbrace{\sum_{i=1}^{m} \sum_{t \in \mathcal{I}_i} \mathbb{E}_{\tilde{x}_{j,x,a}^t} \left[ \log \left( \frac{\hat{p}_j^{t,t_i}(\tilde{x}_{j,x,a}^t|x,a) + \frac{1}{|\mathcal{X}|}}{2} \right) - \log \left( \frac{\hat{p}_j^t(\tilde{x}_{j,x,a}^t|x,a) + \frac{1}{|\mathcal{X}|}}{2} \right) \right]}_{(ii)},
$$

where the expectation of the second term is with respect to $\tilde{x}_{j,x,a}^t \sim \left( p_j^t(\cdot|x,a) + \frac{1}{|\mathcal{X}|} \right)/2$.

We analyse each term separately:

First, note that $(ii)$ is the expectation over $\tilde{x}_{j,x,a}^t$ of the cumulative regret of sleeping EWA on interval $\mathcal{I}_i$ with respect to the expert $t_i$ using the loss function $\ell^t$ defined in Eq. (12). This term is upper bounded by $\log(T)$ (see Subsection 4.3 of the main paper). From it we deduce that

$$
(ii) \leq \sum_{i=1}^{m} \log(T) = m \log(T).
$$

Regarding term $(i)$, we start by using the inverse of Pinsker's inequality presented in Lemma C.4,

$$
(i) \leq \sum_{i=1}^{m} \sum_{t \in \mathcal{I}_i} 2|\mathcal{X}| \left\| \frac{p_j^t(\cdot|x,a) + \frac{1}{|\mathcal{X}|}}{2} - \left( \frac{\hat{p}_j^{t,t_i}(\cdot|x,a) + \frac{1}{|\mathcal{X}|}}{2} \right) \right\|_1^2
$$
$$
= \sum_{i=1}^{m} \sum_{t \in \mathcal{I}_i} \frac{|\mathcal{X}|}{2} \|p_j^t(\cdot|x,a) - \hat{p}_j^{t,t_i}(\cdot|x,a)\|_1^2.
$$

To simplify notations, from now on we let $C := \sqrt{\frac{1}{2} \log \left( \frac{N|\mathcal{X}||\mathcal{A}|2^{|\mathcal{X}|}T}{\delta} \right)}$. Applying Lemma C.3, we obtain that

$$
\sum_{i=1}^{m} \sum_{t \in \mathcal{I}_i} \frac{|\mathcal{X}|}{2} \|p_j^t(\cdot|x,a) - \hat{p}_j^{t,t_i}(\cdot|x,a)\|_1^2
$$
$$
\leq \frac{|\mathcal{X}|}{2} \sum_{i=1}^{m} \sum_{t \in \mathcal{I}_i} \left[ \frac{C^2}{t - t_i + 1} + \frac{2C}{\sqrt{t - t_i + 1}} \sum_{k=t_i}^{t-1} \Delta_k^p + \left( \sum_{k=t_i}^{t-1} \Delta_k^p \right)^2 \right]
$$
$$
\leq \frac{|\mathcal{X}|}{2} \sum_{i=1}^{m} \left[ C^2 \log(|\mathcal{I}_i|) + 2C\sqrt{|\mathcal{I}_i|}\Delta_{\mathcal{I}_i}^p + |\mathcal{I}_i|(\Delta_{\mathcal{I}_i}^p)^2 \right]
$$
$$
\leq \frac{|\mathcal{X}|}{2} \left[ C^2 m \log(T) + 2C\sqrt{T}\frac{\Delta^p}{\sqrt{m}} + T\frac{(\Delta^p)^2}{m^2} \right].
$$

Joining the upper bounds of $(i)$ and $(ii)$ we conclude that

$$
\sum_{t=1}^{T} \|p_j^t(\cdot|x,a) - \hat{p}_j^t(\cdot|x,a)\|_1^2 \leq 8\big((i) + (ii)\big)
$$
$$
\leq 8\Big( (C^2\frac{|\mathcal{X}|}{2} + 1)m\log(T) + 2\frac{|\mathcal{X}|}{2}C\sqrt{T}\frac{\Delta^p}{\sqrt{m}} + T\frac{|\mathcal{X}|}{2}\frac{(\Delta^p)^2}{m^2} \Big).
$$

Thus, for $m = \left( \frac{2T\Delta^p}{C^2 \log(T)} \right)^{1/3}$,

$$
\sum_{t=1}^{T} \|p_j^t(\cdot|x,a) - \hat{p}_j^t(\cdot|x,a)\|_1^2 \leq 12|\mathcal{X}|C^{4/3}\log(T)^{2/3}T^{1/3}(\Delta^p)^{2/3}. \tag{17}
$$

**Back to the analysis of $R_{[T]}^{\hat{p}}$.** Using the inequality in Eq. (17) to bound the $L_1$ norm of $(*)$ on Eq. (16), we obtain that

$$
R_{[T]}^{\hat{p}} \leq L_F N\sqrt{T}\sqrt{\sum_{n=1}^{N} \sum_{j=1}^{n-1} \sum_{x,a} 12|\mathcal{X}|C^{4/3}\log(T)^{2/3}T^{1/3}(\Delta^p)^{2/3}}
$$
$$
\leq 2L_F N^2|\mathcal{X}|\sqrt{3|\mathcal{A}|}C^{2/3}\log(T)^{1/3}T^{2/3}(\Delta^p)^{1/3},
$$

concluding the proof. $\qquad \square$

Note that for $\sum_{i=1}^{m}\sum_{t\in\mathcal{I}_i} R_{\mathcal{I}_i}^{\hat{p}}(\pi^{t,t_i,\lambda})$ (the third term of the meta-regret decomposition of Eq. (13)), following the same procedure as above, we obtain that

$$\sum_{i=1}^{m}\sum_{t\in\mathcal{I}_i} F^t(\mu^{\pi^{t,t_i},\hat{p}^t}) - F^t(\mu^{\pi^{t,t_i},p^t})$$

$$\leq \sum_{i=1}^{m}\sum_{t\in\mathcal{I}_i} \langle \nabla F^t(\mu^{\pi^{t,t_i},\hat{p}^t}), \mu^{\pi^{t,t_i},\hat{p}^t} - \mu^{\pi^{t,t_i},p^t}\rangle$$

$$\leq L_F \sum_{i=1}^{m}\sum_{t\in\mathcal{I}_i} \|\mu_n^{\pi^{t,t_i},\hat{p}^t} - \mu_n^{\pi^{t,t_i},p^t}\|_1$$

$$\leq L_F \sum_{i=1}^{m}\sum_{t\in\mathcal{I}_i}\sum_{n=1}^{N}\sum_{j=1}^{n}\sum_{x,a} \mu_{j-1}^{\pi^{t,t_i},p^t}(x,a)\|p_j^t(\cdot|x,a) - \hat{p}_j^t(\cdot|x,a)\|_1$$

$$\leq L_F \sqrt{\underbrace{\sum_{i=1}^{m}\sum_{t\in\mathcal{I}_i}\sum_{n=1}^{N}\sum_{j=1}^{n}\sum_{x,a}(\mu_{j-1}^{\pi^{t,t_i},p^t}(x,a))^2}_{\leq TN^2} \underbrace{\sum_{t=1}^{T}\sum_{n=1}^{N}\sum_{j=1}^{n}\sum_{x,a}\|p_j^t(\cdot|x,a) - \hat{p}_j^t(\cdot|x,a)\|_1^2}_{\text{independent of } \pi^{t,t_i}}}.$$

Since the second term is independent of $\pi^{t,t_i}$, the analysis is the same as before and we obtain the same upper bound as for $R_{[T]}^{\hat{p}}(\pi^t)$.

# E   Proof of Prop. 5.4: $R_{[T]}^{\text{black-box}}$ regret analysis

*Proof.* Assume a Black-box algorithm satisfying the dynamic regret bound of Eq. (10), i.e., for any interval $I \subseteq [T]$, with respect to any sequence of policies $(\pi^{t,*})_{t\in I}$, and for any learning rate $\lambda$,

$$R_I\big((\pi^{t,*})_{t\in I}\big) \leq c_1\lambda|I| + \frac{c_2\Delta_I^{\pi^*} + c_3}{\lambda} + |I|\Delta_I^p. \tag{18}$$

Consider any sequence of episodes $1 = t_1 < t_2 < \ldots < t_{m+1} = T+1$, forming intervals $\mathcal{I}_i := [t_i, t_{i+1} - 1]$ for all $i \in [m]$. We can decompose the black-box regret over $[T]$ as

$$R_{[T]}^{\text{black-box}}\big((\pi^{t,*})_{t\in[T]}\big) = \sum_{i=1}^{m}\sum_{t\in\mathcal{I}_i} F^t(\mu^{\pi^{t,t_i,\lambda},p^t}) - F^t(\mu^{\pi^{t,*},p^t})$$

$$\leq \sum_{i=1}^{m} c_1\lambda|\mathcal{I}_i| + \frac{c_2\Delta_{\mathcal{I}_i}^{\pi^*} + c_3}{\lambda} + |\mathcal{I}_i|\Delta_I^p$$

$$\leq c_1\lambda T + \frac{c_2\Delta^{\pi^*} + c_3}{\lambda} + \frac{T\Delta^p}{m}.$$

In principle, we would like to select the optimal $\lambda$ that optimizes this dynamic regret. However, as $\Delta^{\pi^*}$ may be unknown in advance, this is not possible. We show here that running MetaCURL with the learning rate set $\Lambda := \{2^{-j}|j = 0, 1, 2, \ldots, \lceil\log_2(T)/2\rceil\}$ ensures that the optimal empirical learning rate is close to the true optimal one by a factor of 2 and that the learner always plays as well as the optimal empirical learning rate.

Denote by $\lambda^*$ the optimal learning rate and $\hat{\lambda}^*$ the empirical optimal learning rate in $\Lambda$. Note that

$$\lambda^* = \sqrt{\frac{c_2\Delta^{\pi^*} + c_3}{c_1 T}}.$$

We consider three different cases:

**If $\lambda^* \geq 1$:** this implies that $\frac{c_2\Delta^{\pi^*}+c_3}{c_1 T} \geq 1$. Therefore, we have that $T \leq \frac{c_2\Delta^{\pi^*}+c_3}{c_1}$. As we assume $f_n^t \in [0,1]$ for all time steps $n \in [N]$ and episodes $t \in [T]$, then

$$R_{[T]}^{\text{black-box}}\big((\pi^{t,*})_{t\in[T]}\big) \leq NT + \frac{T\Delta^p}{m} \leq N\left(\frac{c_2\Delta^{\pi^*} + c_3}{c_1}\right) + \frac{T\Delta^p}{m}.$$

**If $\lambda^* \leq 1/\sqrt{T}$:** this implies that $\frac{c_2\Delta^{\pi^*}+c_3}{c_1} \leq 1$. Therefore, taking $\hat{\lambda}^* = 1/\sqrt{T} \in \Lambda$, we have that

$$R_{[T]}^{\text{black-box}}\big((\pi^{t,*})_{t\in[T]}\big) \leq \lambda c_1 T + \frac{c_1}{\lambda} + \frac{T\Delta^p}{m} \leq c_1\sqrt{T} + \frac{T\Delta^p}{m}$$

**If $\lambda^* \in [1/\sqrt{T}, 1]$:** in this case, given the construction of $\Lambda$, there is a $\hat{\lambda}^* \in \Lambda$ such that $\lambda^* \leq \hat{\lambda}^* \leq 2\lambda^*$. Hence,

$$R_{[T]}^{\text{black-box}}\big((\pi^{t,*})_{t\in[T]}\big) \leq 3\sqrt{c_1(c_2\Delta^{\pi^*} + c_3 m)T} + \frac{T\Delta^p}{m}.$$

Therefore, by taking $\lambda = \hat{\lambda}^*$ in the analysis, we can ensure that

$$R_{[T]}^{\text{black-box}}\big((\pi^{t,*})_{t\in[T]}\big) = \sum_{i=1}^{m}\sum_{t\in\mathcal{I}_i} F^t(\mu^{\pi^{t,t_i,\lambda},p^t}) - F^t(\mu^{\pi^{t,*},p^t})$$

$$\leq N\left(\frac{c_2\Delta^{\pi^*} + c_3}{c_1}\right) + c_1\sqrt{T} + 3\sqrt{c_1(c_2\Delta^{\pi^*} + c_3 m)T} + \frac{T\Delta^p}{m}.$$

$\square$

# F  Proof of Thm. 5.1: Main result

Joining the results from the meta-regret bound and the black-box regret bound, we get the main result of the paper:

**Theorem** (Main result). *Let $\delta \in (0,1)$. Playing MetaCURL, with black-box algorithm with dynamic regret as in Eq.* (10)*, with a learning rate grid $\Lambda := \left\{2^{-j}\,|\,j = 0, 1, 2, \ldots, \lceil 1/2\log_2(T)\rceil\right\}$, and with EWA as the sleeping expert subroutine, we obtain, with probability at least $1 - 2\delta$, for any sequence of policies $(\pi^{t,*})_{t\in[T]}$,*

$$R_{[T]}\big((\pi^{t,*})_{t\in[T]}\big) \leq \tilde{O}\Big(\sqrt{\Delta^{\pi^*}T} + \min\big\{\sqrt{T\Delta_\infty^p},\ T^{2/3}(\Delta^p)^{1/3}\big\}\Big).$$

*Proof.* Define a sequence of episodes $1 = t_1 < t_2 < \ldots < t_{m+1} = T+1$, with $\mathcal{I}_i := [t_i, t_{i+1} - 1]$, such that $\Delta_{\mathcal{I}_i}^p \leq \Delta^p/m$ for all $i \in [m]$.

The dynamic regret of $\mathcal{M}(\mathcal{E}, \Lambda)$ with respect to any sequence of policies $(\pi^{t,*})_{t\in[T]}$, and any $\lambda \in \Lambda$, can be decomposed as

$$R_{[T]}\big((\pi^{t,*})_{t\in[T]}\big) = \underbrace{\sum_{i=1}^{m}\sum_{t\in\mathcal{I}_i} F^t(\mu^{\pi^t,p^t}) - F^t(\mu^{\pi^{t,t_i,\lambda},p^t})}_{\text{Meta algorithm regret}} + \underbrace{\sum_{i=1}^{m}\sum_{t\in\mathcal{I}_i} F^t(\mu^{\pi^{t,t_i,\lambda},p^t}) - F^t(\mu^{\pi^{t,*},p^t})}_{\text{Black-box regret on }\mathcal{I}_i}.$$

$$:= R_{[T]}^{\text{meta}} + R_{[T]}^{\text{black-box}}\big((\pi^{t,*})_{t\in[T]}\big).$$

From Prop. 5.3, we have that with probability at least $1 - 2\delta$, and $C := \sqrt{\frac{1}{2}\log\left(\frac{N|\mathcal{X}||\mathcal{A}|2^{|\mathcal{X}|}T}{\delta}\right)}$,

$$R_{[T]}^{\text{meta}} \leq 4L_F N^2|\mathcal{X}|\sqrt{3|\mathcal{A}|}C^{2/3}\log(T)^{1/3}T^{2/3}(\Delta^p)^{1/3} + \sqrt{\frac{mT}{2}\log(T|\Lambda|)}.$$

In addition, for $\Lambda := \left\{2^{-j}\,|\,j = 0, 1, 2, \ldots, \lceil 1/2\log_2(T)\rceil\right\}$, and $\lambda$ equal the best empirical learning rate in $\Lambda$, Prop. 5.4 yields that, if the black-box algorithm has dynamic regret as in Eq. (10) for any interval in $T$, then

$$R_{[T]}^{\text{black-box}}\big((\pi^{t,*})_{t\in[T]}\big) \leq N\left(\frac{c_2\Delta^{\pi^*} + c_3}{c_1}\right) + c_1\sqrt{T} + 3\sqrt{c_1(c_2\Delta^{\pi^*} + c_3 m)T} + \frac{T\Delta^p}{m}.$$

Therefore, joining both results, we get that,

$$R_{[T]}\big((\pi^{t,*})_{t\in[T]}\big) \leq 4L_F N^2 |\mathcal{X}|\sqrt{3|\mathcal{A}|}C^{2/3}\log(T)^{1/3}T^{2/3}(\Delta^p)^{1/3} + \sqrt{\frac{mT}{2}\log(T\log_2(T))}$$

$$+ N\left(\frac{c_2\Delta^{\pi^*}+c_3}{c_1}\right) + c_1\sqrt{T} + 3\sqrt{c_1(c_2\Delta^{\pi^*}+c_3 m)T} + \frac{T\Delta^p}{m}.$$

Thus, for $m = \left(\frac{2\sqrt{T}\Delta^p}{\gamma}\right)^{2/3}$, with $\gamma := \sqrt{\frac{\log(T\log_2(T))}{2}} + 3\sqrt{c_1 c_3}$, we have that

$$R_{[T]}\big((\pi^{t,*})_{t\in[T]}\big) \leq T^{2/3}\Delta^{1/3}\big(4L_F N^2|\mathcal{X}|\sqrt{3|\mathcal{A}|}C^{2/3}\log(T)^{1/3} + 2\gamma^{2/3}\big)$$

$$+ N\left(\frac{c_2\Delta^{\pi^*}+c_3}{c_1}\right) + c_1\sqrt{T} + 3\sqrt{c_1 c_2\Delta^{\pi^*}T}$$

$$= \tilde{O}\left(\sqrt{\Delta^{\pi^*}T} + \min\big\{\sqrt{T\Delta^p_\infty},\ T^{2/3}(\Delta^p)^{1/3}\big\}\right).$$

$\square$

# G  Greedy MD-CURL dynamic regret analysis

Here we introduce Greedy MD-CURL developed by [39], a computationally efficient policy-optimization algorithm known for achieving sublinear static regret in online CURL with adversarial objective functions within a stationary MDP. We begin by detailing Greedy MD-CURL as presented in [39] in Alg. 4. We then provide a new analysis upper bounding the dynamic regret of Greedy MD-CURL in a quasi-stationary interval valid for any learning rate $\lambda$. Hence, Greedy MD-CURL can be used as a black-box for MetaCURL. This is the first dynamic regret analysis for a CURL approach.

Let $\mathcal{M}^{p,*}_{\mu_0}$ denote the subset of $\mathcal{M}^p_{\mu_0}$ where the corresponding policies $\pi$ are such that $\pi_n(a|x) \neq 0$ for all $(x,a) \in \mathcal{X} \times \mathcal{A}$, $n \in [N]$. For any probability transition $p$, $\Gamma : \mathcal{M}^p_{\mu_0} \times \mathcal{M}^{p,*}_{\mu_0} \to \mathbb{R}$ such that, for all $\mu \in \mathcal{M}^p_{\mu_0}$ with its associated policy $\pi$, and $\mu' \in \mathcal{M}^{p,*}_{\mu_0}$ with its associated policy $\pi'$, we have

$$\Gamma(\mu^\pi, \mu^{\pi'}) := \sum_{n=1}^{N} \mathbb{E}_{(x,a)\sim\mu^\pi_n(\cdot)}\left[\log\left(\frac{\pi_n(a|x)}{\pi'_n(a|x)}\right)\right]. \tag{19}$$

This divergence $\Gamma$ is a Bregman divergence (see Proposition 4.3 of [39]). Problem (20) implemented with this Bregman divergence $\Gamma$ has a closed form solution, as showed in [39].

**Algorithm 4** Greedy MD-CURL

---

1: **Input:** number of episodes $T$, initial sequence of policies $\pi^1 \in (\mathcal{S}_\mathcal{A})^{\mathcal{X} \times N}$, initial state-action distribution $\mu_0$, learning rate $\lambda > 0$, sequence of parameters $(\alpha_t)_{t \in [T]}$.

2: **Initialization:** $\quad \forall (x, a), \hat{p}^1(\cdot | x, a) = \frac{1}{|\mathcal{X}|}$ and $\mu^1 = \tilde{\mu}^1 := \mu^{\pi^1, \hat{p}^1}$

3: **for** $t = 1, \ldots, T$ **do**

4: $\quad$ Agent starts at $(x_0^t, a_0^t) \sim \mu_0(\cdot)$

5: $\quad$ **for** $n = 1, \ldots, N$ **do**

6: $\qquad$ Environment draws new state $x_n^t \sim p_n(\cdot | x_{n-1}^t, a_{n-1}^t)$

7: $\qquad$ Learner observes agent's external noise $\varepsilon_n^t$

8: $\qquad$ Agent chooses an action $a_n^t \sim \pi_n^t(\cdot | x_n^t)$

9: $\quad$ **end for**

10: $\quad$ Update probability kernel estimate for all $(x, a)$:

11:
$$\hat{p}_n^{t+1}(\cdot | x, a) := \frac{1}{t} \delta_{g_n(x, a, \epsilon_n^t)} + \frac{t-1}{t} \hat{p}_n^t(\cdot | x, a)$$

12: $\quad$ Compute policy for the next episode:

13:
$$\mu^{t+1} \in \underset{\mu \in \mathcal{M}_{\mu_0}^{\hat{p}^{t+1}}}{\arg \min} \{ \lambda \langle \nabla F^t(\mu^t), \mu \rangle + \Gamma(\mu, \tilde{\mu}^t) \} \tag{20}$$

14: $\quad$ for all $n \in [N], (x, a) \in \mathcal{X} \times \mathcal{A}$,

$$\pi_n^{t+1}(a|x) = \frac{\mu_n^{t+1}(x, a)}{\sum_{a' \in \mathcal{A}} \mu_n^{t+1}(x, a')}$$

15: $\quad$ Compute $\tilde{\pi}^{t+1} := (1 - \alpha_{t+1})\pi^{t+1} + \alpha_{t+1}/|\mathcal{A}|$

16: $\quad$ Compute $\tilde{\mu}^{t+1} := \mu^{\tilde{\pi}^{t+1}, \hat{p}^{t+1}}$ as in Eq. (2)

17: **end for**

18: **return** $(\pi^t)_{t \in [T]}$

---

### G.1 Dynamic regret analysis of Greedy MD-CURL

Let us assume we analyze our regret in an interval $I := [t_s, t_e] \subseteq [T]$. We denote by $R_I$ the dynamic regret of an instance of Greedy MD-CURL started at episode $t_s$ until the end of interval $I$ at episode $t_e$. We denote by $\pi^t$ the policy produced by this instance of Greedy MD-CURL at episode $t \in I$, $p^t$ the true probability transition kernel, and

$$\hat{p}_n^t(x'|x, a) = \frac{1}{t - t_s} \sum_{s=t_s}^{t-1} \mathbb{1}_{\{g_n(x, a, \varepsilon_n^s) = x'\}},$$

the empirical estimate of the probability kernel at episode $t$, with data from the beginning of the interval $I$.

We define and decompose the dynamic regret $R_I$ with respect to any sequence of policies $(\pi^{t,*})_{t \in I}$ into three terms as follows:

$$R_I\big((\pi^{t,*})_{t \in I}\big) := \sum_{t \in I} F^t(\mu^{\pi^t, p^t}) - F^t(\mu^{\pi^{*,t}, p^t}) = \underbrace{\sum_{t \in I} F^t(\mu^{\pi^t, p^t}) - F^t(\mu^{\pi^t, \hat{p}^t})}_{R_I^{\text{MDP}}(\pi^t)}$$

$$+ \underbrace{\sum_{t \in I} F^t(\mu^{\pi^t, \hat{p}^t}) - F^t(\mu^{\pi^{*,t}, \hat{p}^t})}_{R_I^{\text{policy}}\big((\pi^{t,*})_{t \in I}\big)} + \underbrace{\sum_{t \in I} F^t(\mu^{\pi^{t,*}, \hat{p}^t}) - F^t(\mu^{\pi^{t,*}, p^t})}_{R_I^{\text{MDP}}(\pi^{t,*})}. \tag{21}$$

The terms $R_I^{\text{MDP}}(\pi^t)$ and $R_I^{\text{MDP}}(\pi^{t,*})$ pay for our lack of knowledge of the true MDP, forcing us to use its empirical estimate. The term $R_I^{\text{policy}}$ corresponds to the loss incurred in calculating the policy

by solving the optimization problem given in Eq. (20). Below, we present the first analysis of the dynamic regret for a CURL algorithm. We consider each term separately.

### G.1.1 $R_I^{\text{MDP}}$ analysis

In Section 2 we assume that the deterministic part of the dynamics, given by $g_n$ in equation (8) for each time step $n$, is known in advance. The source of uncertainty and non-stationarity in the MDP comes only from the external noise dynamics, that is independent of the agent's state-action pair. Therefore, we do not need to explore in this setting, so the analysis of the two terms $R_I^{\text{MDP}}(\pi^t)$ and $R_I^{\text{MDP}}(\pi^{t,*})$ are the same.

**Proposition G.1.** *With probability at least $1 - \delta$,*

$$R_I^{MDP}(\pi^t) \le L_F N^2 \sqrt{\frac{1}{2} \log\left(\frac{N|\mathcal{X}||\mathcal{A}|2^{|\mathcal{X}|}T}{\delta}\right)} \sqrt{|I|} + |I|\Delta_I^p,$$

*for all intervals $I \in [T]$. The same result is valid for $R_I^{MDP}(\pi^{t,*})$.*

*Proof.* We start by using the convexity of $F^t$, Holder's inequality, that $F^t$ is $L_F$-Lipschitz, and Lemma C.5 to obtain that

$$R_I^{\text{MDP}}(\pi^t) \le L_F \sum_{t=t_s}^{t_e} \sum_{n=1}^{N} \sum_{j=1}^{n} \sum_{x,a} \mu_{j-1}^{\pi^{i,t},p^t}(x,a) \|p_j^t(\cdot|x,a) - \hat{p}_j^t(\cdot|x,a)\|_1. \tag{22}$$

Applying Lemmas C.1 and C.2, we have that for any $\delta \in (0,1)$, with probability $1 - \delta$,

$$\|p_j^t(\cdot|x,a) - \hat{p}_j^t(\cdot|x,a)\|_1 \le \|p_j^t(\cdot|x,a) - \overline{p}_j^t(\cdot|x,a)\|_1 + \|\overline{p}_j^t(\cdot|x,a) - \hat{p}_j^t(\cdot|x,a)\|_1$$

$$\le \sqrt{\frac{1}{2(t - t_s)} \log\left(\frac{N|\mathcal{X}||\mathcal{A}|2^{|\mathcal{X}|}T}{\delta}\right)} + \sum_{k=t_s}^{t-1} \Delta_k^p.$$

Using this to continue the upper bound of Eq. (22), we conclude our proof:

$$R_I^{\text{MDP}}(\pi^t) \le L_F N^2 \sum_{t=t_s}^{t_e} \left(\sqrt{\frac{1}{2(t - t_s)} \log\left(\frac{N|\mathcal{X}||\mathcal{A}|2^{|\mathcal{X}|}T}{\delta}\right)} + \sum_{k=t_s}^{t-1} \Delta_k^p\right)$$

$$\le L_F N^2 \sqrt{\frac{1}{2} \log\left(\frac{N|\mathcal{X}||\mathcal{A}|2^{|\mathcal{X}|}T}{\delta}\right)} \sqrt{|I|} + |I|\Delta_I^p.$$

$\square$

### G.1.2 $R_I^{\text{policy}}$ analysis

**Proposition G.2.** *Let $b$ be a constant defined as*

$$b := 8N^2 + 2N^2 \log(|\mathcal{A}|) \log(|I|)\big(1 + \log(|I|)\big) + 2N \log(|I|) + N \log(|\mathcal{A}|).$$

*Then, Greedy MD-CURL obtains, for any sequence of policies $(\pi^{t,*})_{t \in I}$, and for any learning rate $\lambda > 0$,*

$$R_I^{policy}\big((\pi^{t,*})_{t \in I}\big) \le \lambda L_F^2 |I| + \frac{N^2}{\lambda} \Delta_I^{\pi^*} + b\frac{1}{\lambda}.$$

*Proof.* We adapt the proof of Prop. 5.7 of [39] that upper bounds the static regret incurred when solving the optimization Problem (20), for a proof that upper bounds the dynamic regret. The main difference is that, in the case of static regret, we compare ourselves to the same policy throughout the interval, whereas in the case of dynamic regret, at each episode we compare ourselves to a different policy given by $\pi^{t,*}$. Consequently, the analysis remains the same as in [39] for all terms that do not depend on $\pi^{t,*}$, but requires a new analysis in terms that do depend on it.

To simplify notation, we take $\ell^t := \nabla F^t(\mu^{\pi^t,\hat{p}^t})$ and $\mu^t := \mu^{\pi^t,\hat{p}^t}$. We can use the same reasoning as in appendix $D.5$ of [39] to show that

$$R_I^{\text{policy}} \leq \underbrace{\frac{1}{\lambda}\sum_{t\in I}\left[\lambda\langle\ell^t,\mu^t-\mu^{t+1}\rangle - \Gamma(\mu^{t+1},\tilde{\mu}^t)\right]}_{A} + \underbrace{\frac{1}{\lambda}\sum_{t\in I}\left[\Gamma(\mu^{\pi^{t,*},\hat{p}^{t+1}},\tilde{\mu}^t) - \Gamma(\mu^{\pi^{t,*},\hat{p}^{t+1}},\mu^{t+1})\right]}_{B}.$$

Since the $A$ term does not depend on $\pi^{t,*}$, its analysis follows directly from [39], and is given by

$$A \leq \lambda L_F^2|I| + \frac{1}{2\lambda}\sum_{t\in I}\left(\frac{2N}{t-t_s}+2N\alpha_t\right)^2, \tag{23}$$

where $L_F$ is the Lipschitz constant of $F^t$ and $\alpha^t$ is an input parameter of Greedy MD-CURL.

We then proceed to analyze term $B$. Again, following the procedure of appendix $D.5$ of [39], we obtain that

$$B = \underbrace{\sum_{t=1}^T \Gamma(\mu^{\pi^{t,*},\hat{p}^{t+1}},\tilde{\mu}^t) - \Gamma(\mu^{\pi^{t-1,*},\hat{p}^t},\tilde{\mu}^t)}_{(i)} + \underbrace{\sum_{t=1}^T \Gamma(\mu^{\pi^{t-1,*},\hat{p}^t},\tilde{\mu}^t) - \Gamma(\mu^{\pi^{t-1,*},\hat{p}^t},\mu^t)}_{(ii)}$$

$$+ \underbrace{\sum_{t=1}^T \Gamma(\mu^{\pi^{t-1,*},\hat{p}^t},\mu^t) - \Gamma(\mu^{\pi^{t,*},\hat{p}^{t+1}},\mu^{t+1})}_{(iii)}.$$

Let $\psi : (\mathcal{S}_{\mathcal{X}\times\mathcal{A}})^N \to \mathbb{R}$ denote the function inducing the Bregman divergence $\Gamma$ of Eq. (19). [39] further shows that:

- $(i) \leq -\psi(\mu^{\pi^{t_i-1,*},\hat{p}^{t_i}}) + N\sum_{t\in I}\log\left(\frac{|\mathcal{A}|}{\alpha_t}\right)\|\mu^{\pi^{t-1,*},\hat{p}^t}-\mu^{\pi^{t,*},\hat{p}^{t+1}}\|_{\infty,1}$

- $(ii) \leq 2N\sum_{t\in I}\alpha_t$, and this upper bound is found independently of $\pi^{t,*}$

- $(iii) \leq \Gamma(\mu^{\pi^{t-1,*},\hat{p}^t},\mu^t)$.

Lemma $D.6$ of [39] shows that,

$$-\psi(\mu^{\pi^{t_i-1,*},\hat{p}^{t_i}}) + \Gamma(\mu^{\pi^{t_i-1,*},\hat{p}^{t_i}},\mu^{t_i}) \leq N\log(|\mathcal{A}|).$$

Only term $(i)$, which involves $\|\mu^{\pi^{*,t-1},\hat{p}^t}-\mu^{\pi^{*,t},\hat{p}^{t+1}}\|_{\infty,1}$, depends on the sequence $(\pi^{t,*})_{t\in[T]}$, requiring then a new analysis. For this purpose, we rely on the following two results:

- From Lemma 5.6 of [39], we have that, for all strategies $\pi$,

$$\|\mu^{\pi,\hat{p}^t}-\mu^{\pi,\hat{p}^{t+1}}\|_{\infty,1} \leq \frac{2N}{t-t_s}.$$

- From auxiliary Lemma C.6 proved in Appendix C we have that

$$\|\mu_n^{\pi^{*,t-1},\hat{p}^{t+1}}-\mu_n^{\pi^{t,*},\hat{p}^{t+1}}\|_1 \leq \sum_{i=1}^n\sum_{x\in\mathcal{X}}\rho_i^{\pi^{t-1,*}}(x)\|\pi_i^{t,*}(\cdot|x)-\pi_i^{t-1,*}\|_1 \leq N\Delta_t^{\pi^*}.$$

Therefore, using the triangular inequality and the two results above, we obtain that

$$\|\mu^{\pi^{*,t-1},\hat{p}^t}-\mu^{\pi^{t,*},\hat{p}^{t+1}}\|_{\infty,1} \leq \|\mu^{\pi^{t-1,*},\hat{p}^t}-\mu^{\pi^{t-1,*},\hat{p}^{t+1}}\|_{\infty,1} + \|\mu^{\pi^{t-1,*},\hat{p}^{t+1}}-\mu^{\pi^{t,*},\hat{p}^{t+1}}\|_{\infty,1}$$

$$\leq \frac{2N}{t} + N\Delta_t^{\pi^*}.$$

Therefore, the bound on term $B$ is given by

$$B \leq \frac{1}{\lambda}\left[N\sum_{t\in I}\log\left(\frac{|\mathcal{A}|}{\alpha_t}\right)\left(\frac{2N}{t-t_s}+N\Delta_t^{\pi^*}\right) + 2N\sum_{t\in I}\alpha_t + N\log(|\mathcal{A}|)\right]. \tag{24}$$

**Final step: joining all results**

Joining the upper bounds on term $A$ from Eq. (23) and on term $B$ from Eq. (24), we have that

$$R_I^{\text{policy}}\big((\pi^{t,*})_{t\in I}\big) \le A + B$$

$$\le \lambda L_F^2|I| + \frac{1}{2\lambda}\sum_{t\in I}\left(\frac{2N}{t-t_s} + 2N\alpha_t\right)^2$$

$$+ \frac{1}{\lambda}\left[N\sum_{t\in I}\log\left(\frac{|\mathcal{A}|}{\alpha_t}\right)\left(\frac{2N}{t-t_s} + N\Delta_t^{\pi^*}\right) + 2N\sum_{t\in I}\alpha_t + N\log(|\mathcal{A}|)\right].$$

If we take the learning rate as $\alpha_t = 1/t$, then, for all $\lambda > 0$,

$$R_I^{\text{policy}}\big((\pi^{t,*})t\in I\big) \le \lambda L_F^2|I| + \frac{N^2}{\lambda}\Delta_I^{\pi^*}$$

$$+ \frac{1}{\lambda}\left[8N^2 + 2N^2\log(|\mathcal{A}|)\log(|I|)\big(1 + \log(|I|)\big) + 2N\log(|I|) + N\log(|\mathcal{A}|)\right]$$

$$= \lambda L_F^2|I| + \frac{N^2\Delta_I^{\pi^*} + b}{\lambda},$$

where $b := 8N^2 + 2N^2\log(|\mathcal{A}|)\log(|I|)\big(1 + \log(|I|)\big) + 2N\log(|I|) + N\log(|\mathcal{A}|)$.

$\square$

## G.2 Final Greedy MD-CURL regret analysis

Replacing the bounds of Prop. G.1 and G.2 in Eq. (21) yields the final upper bound of Greedy MD-CURL's dynamic regret for any interval $I \subseteq T$ with respect to any sequence of policies $(\pi^{t,*})_{t\in I}$:

**Theorem G.3** (Dynamic regret of Greedy MD-CURL). *Let $b$ be a constant defined as*

$$b := 8N^2 + 2N^2\log(|\mathcal{A}|)\log(|I|)\big(1 + \log(|I|)\big) + 2N\log(|I|) + N\log(|\mathcal{A}|).$$

*Let $\delta \in (0,1)$. With probability at least $1 - 2\delta$, for any interval $I \subseteq [T]$, for any sequence of policies $(\pi^{t,*})_{t\in I}$, for any learning rate $\lambda > 0$, and for $\alpha_t := 1/t$, Greedy MD-CURL obtains*

$$R_I\big((\pi^{t,*})_{t\in I}\big) \le \lambda L_F^2|I| + \frac{N^2\Delta_I^{\pi^*} + b}{\lambda} + 2FN^2\sqrt{\frac{1}{2}\log\left(\frac{N|\mathcal{X}||\mathcal{A}|2^{|\mathcal{X}|}T}{\delta}\right)}\sqrt{|I|} + 2|I|\Delta_I^p$$

Hence, Greedy MD-CURL meets the requisite dynamic regret bound from Eq. (10) to serve as a black-box algorithm for MetaCURL achieving optimal dynamic regret.

