# OpenReview forum: "MetaCURL: Non-stationary Concave Utility Reinforcement Learning"
_NeurIPS.cc/2024/Conference — NeurIPS 2024 poster_

### Official Review · Reviewer_unHu · 2024-07-08

**Soundness:** 3
**Presentation:** 3
**Contribution:** 2
**Rating:** 6
**Confidence:** 3

**Summary:**

This paper addresses concave utility RL in a non-stationary episodic setting, where the transition probabilities as well as the utility function may change from one episode to the other. The paper proposes an algorithm, dubbed MetaCURL, which dynamically select the best performing "expert" within a set of baseline algorithms with different starting points and hyper-parameters. The paper analyses the dynamic regret of MetaCURL as a function of some characteristics of the instance, such as the number of abrupt changes and the magnitude of changes, under a deterministic dynamics plus action-independent noise assumption.

**Strengths:**

- Novelty of the setting. Concave utility RL has been studied extensively recently, but I am not aware of any work combining concave utilities and non-stationarity;
- Simplicity of the approach. The expert-based solution is neat and easy to follow.

**Weaknesses:**

- Motivation of the setting (1). The paper does not do much to support why studying non-stationarity in CURL is useful/interesting;
- Motivation of the setting (2). The paper only evaluates the performance of the expected realization rather than the expected performance;
- Restrictive assumptions. The setting presents several challenges but does not require exploration due to an assumption that makes the stochasticity of the transitions independent from the actions;
- Analysis. It is unclear whether the analysis brings some interesting techniques beyond what has been done in prior works.

This paper tackles a setting comprising an eye-popping list of challenges, which include: non-stationarity of transitions, adversarial learning, concave cost functions. However, aside from restrictive assumptions that makes strategic exploration unnecessary, I believe the paper somewhat fails in two aspects: First, to motivate the setting properly, not just as a "patchwork" of other settings considered in the literature but as a problem raising from important application; Second, to highlight what is the value of their results in terms of insights and employed techniques.
To these reasons, I am currently providing a slightly negative evaluation. I would like to hear from the authors on these concerns. More details below.

**Questions:**

MOTIVATION: NON-STATIONARY CURL.

Both CURL and non-stationary MDPs are interesting and worth studying, but why the combination of CURL and non-stationarity is interesting? Does the combination brings more challenges than just the sum of its components? Does the algorithm bring novel ideas? Does the analysis bring novel techniques or valuable insights? I feel that the paper shall do more to support why the reported results matter.

MOTIVATION: OCCUPANCY VS SINGLE-TRIAL.

As it is typical in the original formulation, the paper defines the CURL objective as a concave function of the occupancy measure, i.e., the expected state visitations. More recently (see Mutti et al., Challenging common assumptions in convex reinforcement learning, 2022), it has been shown that there is a crucial difference between optimizing the concave objective over the expected visitation and the expected concave objective. Can the authors comment on why the occupancy formulation is relevant in this setting?

MOTIVATION: NO EXPLORATION.

Studying the regret in a setting in which strategic exploration is unnecessary looks a bit odd. Also, the provided motivation is that applying optimism would induce computational issues, which I do not find fully convincing. There exist tractable no-regret algorithms in RL and it shall be shown why it is not the case for CURL. Moreover, plenty of works in RL studies the regret by relying on planning oracles that are intractable to implement. This is a common choice to isolate statistical complexity from computational considerations. I think that considering $g_n$ unknown but belonging to a known function class would make for a significantly more interesting analysis.

NOVELTY OF THE ANALYSIS.

Can the authors highlight what is novel in the analysis? From reading the paper, the main effort seems to reduce the analysis to a combination of EWA and a known rate for the baseline algorithm.

ADDITIONAL REFERENCES.

While the work does a good job of placing the contribution with respect to previous literature in non-stationary RL, there are some missing works that seem relevant. As mentioned above, the papers *(Mutti et al., Challenging common assumptions in convex reinforcement learning, 2022; Mutti et al., Convex reinforcement learning in finite-trials, 2023)* study a so-called "single-trial" variation of the CURL formulation. The works *(Cheung, Regret minimization for reinforcement learning with vectorial feedback and complex objectives, 2019; Cheung, Exploration-exploitation trade-off in reinforcement learning on online markov decision processes with global concave rewards, 2019)* analyse regret-minimization in a problem formulation akin to an infinite-horizon form of CURL. The papers *(Prajapat et al., Submodular reinforcement learning, 2023; De Santi et al., Global reinforcement learning, 2024)* look also closely related.

OTHER COMMENTS.
- Examples of applications: the paper makes an effort of describing a few applications that fulfil the dynamics assumption, not why they are  relevant for CURL.
- Does the analysis need $F_t$ to be fully revealed instead of bandit feedback? If that is the case, can the authors comment on how this assumption may be overcome?
- The considered policies seem to depend only on the current state of the environment (aka Markovian). Can the author discuss why history-based policies are unnecessary in this setting?

**Limitations:**

The paper is mostly upfront in mentioning the limitations introduced by restrictive assumptions.

---

> ### Author Rebuttal · Authors · 2024-08-06
>
> We thank the reviewer for their insightful comments. Below, we address the main concerns:
>
> - **Motivating the Setting:**  In our common response to all reviewers we further elaborate on how the applications presented in the paper align with CURL. We plan to include them earlier in the introduction for the extended version.
>
> - **Novelty of the analysis:** Our approach goes beyond simply combining RL with EWA. The challenge is that, due to the uncertainty and non-stationarity of the environment, the losses of each expert are unknown.
>
> 	Indeed, to use the learning with experts framework we need to estimate the losses of non-played expert policies based solely on the observations from the played policy, given the incomplete knowledge of dynamics. For that, we must construct an estimate $\hat{p}^t$ of the non-stationary probability transition. Using the empirical estimator for $\hat{p}^t$ and standard RL results for bounding the $L_1$ norm between $\hat{p}^t$ and the true dynamics $p^t$ [Neu et al. 2012; 38] cannot be applied due to non-stationarity. To overcome this, we propose a second sleeping expert scheme to compute $\hat{p}^t$ (Alg. 3), where each expert is an empirical estimation of $p^t$ using values from different intervals. Obtaining the optimal rate for it was challenging and required new technical approaches, including a new specific loss function (see Eq. (12) and Alg. 3), and a new regret analysis (see Prop. 5.2).
>
> 	Finally, we need to estimate $\hat{p}^t$ because we are dealing with CURL instead of RL. In RL, the linearity allows us to accurately estimate the loss of each expert's policy, even if the policy is not played, by simulating trajectories with $g\_n$ using data from the played policy. However, in the convex case, this approach does not work, so we must estimate $\hat{p}^t$ to determine the losses of non-played policies.
>
> - **No exploration:** We  address the concern about the dynamic hypothesis in our common answer to all reviewers. We believe our work could be extended to cases where $g_n$ belongs to a known family of parametric functions. We address below the question regarding the computational complexity:
>
> 	 In tabular RL, there are two approaches for dealing with dynamic’s uncertainty and adversarial losses: policy optimization (PO) and occupancy-measure methods. Occupancy-measure methods use ideas from UCRL2 [4]. They construct a set of plausible MDPs compatible with the observed samples and then play the policy that minimizes the cost over this set of plausible MDPs. PO methods evaluates the value function and uses a mirror descent update directly on the policy, solving the optimization problem through dynamic programming to obtain a closed-form solution. PO is then more computationally efficient than planning in the space of all plausible MDPs [Luo et al. 2021, Efroni et al. 2020a,b].
>
> 	No efficient method that fully explores in CURL exists. As PO methods for RL rely on the value function, they are unsuitable for CURL, as CURL invalidates the classical Bellman equations. The occupancy-measure methods can be applied but are computationally less efficient. The algorithm we propose as a black-box is a PO method adapted for CURL from [35], but that also assumes that $ g\_n $ is known.
>
> - **Occupancy vs. single-trial:** Thank you for pointing out the work by Mutti et al., 2022. It is an interesting question and we will discuss it in our paper. We chose to work with the expected realization setting to complement existing CURL research, also benefiting from the algorithm from [35] as a black-box. In scenarios with many homogeneous agents (like those in Section 2), a mean-field approach could justify this choice.
>
> - **Bandit feedback of $F^t$:** This is an interesting question for a future work. The CURL bandit framework is significantly more difficult, in the same way that the convex bandit framework is more difficult than the multi-armed bandit one in online learning. In addition, it is also harder for the meta-algorithm to combine results from bandit algorithms while maintaining the optimal regret rates (see Agarwal et al., 2016).
>
> - **Markovian policies:** In a non-stationary environment history-based policies can be detrimental, as they rely on outdated information. What was learned about the environment in previous episodes might no longer be accurate in the current one. Since the learner is unaware of the variation budget of the environment, the best approach is to discard past knowledge and restart learning at every major change in the environment.
>
>     Moreover, history-based policies are unnecessary for determining optimal restart times. Our meta-algorithm aggregates active instances of a black-box algorithm, ensuring performance at least as good as the best active instance, likely the longest-running one since the last major change in the environment. Given that the environment is near-stationary since the last major change, any history-based policy can be associated with a Markovian policy with the same state-action distribution [Putterman, 1994]. We can then just work with Markovian policies.
>
> - **Additional references:** Thank you for pointing out these interesting new references. We will add them to our related work.
>
>
> ### References:
> - *Neu et. al 2012, The adversarial stochastic shortest path problem with unknown transition probabilities, AISTATS*
> - *Luo et al. 2021, Policy Optimization in Adversarial MDPs: Improved Exploration via Dilated Bonuses, NeurIPS*
> - *Efroni et al. 2020a, Optimistic Policy Optimization with Bandit Feedback, ICML*
> - *Efroni et al. 2020b, Exploration-Exploitation in Constrained MDPs*
> - *Agarwal et. al 2016, Corralling a Band of Bandit Algorithms*
> - *Jin et al. 2020, Learning Adversarial MDPs with Bandit Feedback and Unknown Transition, ICML*

---

> > ### Comment · Reviewer_unHu · 2024-08-10
> >
> > Dear authors,
> >
> > Thank you for your thoughtful replies to my comments and the general response above.
> > I think you are making a compelling enough case for the paper showing why the analysis is interesting and providing some application where the need for models that are non-stationary and concave-utility simultaneously may be reasonable. I am increasing my score towards acceptance. I still believe that sidestepping the exploration problem is not fully motivated and that potential application shall be explored in more details.
> >
> > Some additional comments:
> > - The reported application are interesting, but I want to stress that they sometimes fail to answer the question. For instance, you are explaining why the finance domain may be non-stationary (clear), but not why the utility shall be concave (e.g., through risk aversion);
> > - "No efficient method that fully explores in CURL exists". What about intractable methods that are statistically efficient? Previous work show that CURL can be solved through a sequence of RL problems, so it is not clear what makes CURL inefficient. Is it the instantiation of the RL problem (aka computing the reward function)?

---

> > > ### Author Response · Authors · 2024-08-12
> > >
> > > We would like to thank the reviewer for their prompt response and for recognizing the motivations and novelties in our theoretical analysis. We also appreciate the reviewer's insightful comments and suggestions, and we will incorporate the recommended changes into our paper. Below, we address the reviewer's additional comments.
> > >
> > > > The reported application are interesting, but I want to stress that they sometimes fail to answer the question. For instance, you are explaining why the finance domain may be non-stationary (clear), but not why the utility shall be concave (e.g., through risk aversion);
> > >
> > > We agree that the objective function was not explicitly stated in some cases. In the inventory management domain, for instance, a multi-objective function can be needed, while in finance, the objective might involve minimizing a convex risk measure (e.g., Conditional Value-at-Risk) while maximizing returns. To clarify our motivation, we will carefully explain in the paper how each example provided fully aligns with our framework.
> > >
> > > > “No efficient method that fully explores in CURL exists". What about intractable methods that are statistically efficient? Previous work show that CURL can be solved through a sequence of RL problems, so it is not clear what makes CURL inefficient. Is it the instantiation of the RL problem (aka computing the reward function)?
> > >
> > > We acknowledge that statistically efficient methods for online CURL with adversarial losses exist, such as occupancy-measure approaches using Mirror Descent combined with UCRL2 techniques. However, these methods are not computationally efficient, either for RL or CURL, as they require solving an optimization problem over the set of all plausible MDPs within a confidence interval at each iteration. On the other hand, policy optimization algorithms, which are both statistically and computationally efficient for RL, rely on value-based techniques that cannot be applied to CURL due to the invalidation of the classical Bellman equations.
> > >
> > > The work that demonstrated that CURL can be solved through a sequence of RL problems [Zahavy et al. (2021)], considers a different setting than ours. They consider the CURL problem in Equation (1) with a fixed convex objective function and unknown dynamics. We focus on the online version with adversarial objective functions, i.e. that change with each episode and are unknown to the learner. As a result, their reduction does not apply to our case.
> > >
> > > ### References
> > > - *Zahavy et al. 2021, Reward is Enough for Convex MDPs, NeurIPS*

---

### Official Review · Reviewer_FWrb · 2024-07-12

**Soundness:** 2
**Presentation:** 1
**Contribution:** 3
**Rating:** 4
**Confidence:** 2

**Summary:**

This paper focuses on CURL, i.e., the Concave Utility Reinforcement Learning problem, which can be treated an extension of traditional RL  to deal with convex performance criteria. The authors introduce MetaCURL for non-stationary MDPs. This is a theory heavy paper.

**Strengths:**

Proofs are provided. It seems that the proposed theorems and propositions are novel.

**Weaknesses:**

Actually, for readers (like me) are not an expert in CURL, some introductory examples are helpful. Additionall, it seems that this paper does not have any experiment.

**Questions:**

First of all, I strongly recommend the authors to include at least one or two introductory examples or figures to show the high-level intuition of their settings and motivation. E.g., difference between RL and concave utility RL; a figure showing offline CURL and online CURL; why concave utility RL is important; why do we need to focus on non-stationarity; and so on. Some examples can be within the domains of robotics, medical treatment, and so on.

A good figure example can be found in Fig.1 of [1].

[1] Levine, Sergey, et al. "Offline reinforcement learning: Tutorial, review, and perspectives on open problems." arXiv preprint arXiv:2005.01643 (2020).

A good introductory example can be done by setting some parameters equal to low dimensions.

# Specific Questions about the Settings
1. Can your framework handle cases when the state or the action is continuous?
2. Do we need to assume the occupancy measure is non-zero?

# Inconsistent Notations
1. Authors mention that L247-248, "We assume T experts, with expert s active in interval [s, T]". Actually, such notations are very confusion. $t$ refers to the episode, but here $T$ represents the number of experts. Especially in L209, the authors mention "over a series of T episodes". Again, in L127, the authors mention that "for any interval $I \subseteq [T]$", what interval does $[T]$ represent, $[1:T]$ or $[t:T]$? It is extremely hard for readers to understand their paper with inconsistent notations.

2. In L248, the authors mention that "Expert s at episode t > s outputs". $s$ refers to the expert, how it can be compared with $t$?

3. In L19, authors claim "episodes of length N". Does it mean that each episode has N steps? Do the authors need to assume each episode has the exactly same length?

4. In L248, Why N is treated as a function, $N(x')$? But in L19, authors claim "episodes of length N".

5. In L210, "For each round t". In L225, "In every episode t". Actually, from my perspective, round t is different from episode t.

6. Confusing exponent symbols and notations. In L268, $T^{2/3}$ represents T exponent $2/3$. Is that correct? In your Algorithm 1, symbol $F^{t}$ and $\hat{p}^{t+1}$ represent exponent $t$?

I strongly recommend the authors to introduce a notation table.

# Specific Questions about MetaCURL
As mentioned previously, because the notations are inconsistent, it is really hard for me to fully understand the proposed framework.

1. Learning with Expert Advice (LEA)

    1.1 Do you assume these K experts are optimal? What if there is limited expert knowledge?

    1.2 What if two experts contradict to each other? Will your algorithm take the average of these two values? E.g., for the state $s_1$, expert 1 takes $a_1$, expert_2 takes $-a_1$, What should be the expert loss in that case? Could you show some numerical intro examples?

2. The authors mention that "This problem can be reduced to the sleeping expert problem". However, it is still unclear how and why this problem can be reduced?
    2.1 The authors mention that "experts are not required to provide solutions at every time step". What if at certain step there is not any expert providing solutions? Is there a lower bound of the number of experts that have to be awake?

3. Does the agent have access to the "external noise sequence $\left(\varepsilon_{n}^{t}\right)$"? Because $\epsilon_{n}^{t} \sim h_{n}^{t}(\cdot)$, is there any information leakage?

4. Does the algorithm know which expert is active? Is that signal too strong? Can such information be learned?

5. What is the time complexity of algo. 1?

6. Theorem 5.1

    6.1 In Theorem 5.1, why the complexity is determined by $\Delta^{\pi^{*}}$, $\Delta_{\infty}^{p}$ and $\Delta^{p}$ simultanously? What is the intuition of taking the minimum of abrupt and smooth variation? Under the worst case, should we take the maximum of such variations?

    6.2 In [2], their regret bound contains the dimenions for the state space, action space, and the length of each episode. Why in Theorem 5.1, they are not included? Does it mean that your algorithm will not change, when their dimensions are increasing? Still, because of the inconsistent $T$ notation, it is unclear such $T$ represents? (L247-248, "We assume T experts, with expert s active in interval [s, T]". In L209, the authors mention "over a series of T episodes".)

[2] Rosenberg, Aviv, and Yishay Mansour. "Online convex optimization in adversarial markov decision processes." International Conference on Machine Learning. PMLR, 2019.

7. Because there is not any experiment in the current version, I have general concerns over the practicality of the proposed method, e.g., infinite horizon or continuous spaces.

**Limitations:**

This paper does not have any introductory examples, or figures, or experiments. However, as such a notation dense paper, it seems that the notations are not consistent, making readers extremely hard to understand their major contributions. Additionally, the agents are assumed to have special strong information, e.g., experts signal or external noise sequence, which might be missing in real-world.

-----------------Updates

Thanks for the thoughtful discussions. After careful consideration, I have decided to maintain my current rating of 4.

tl&dr: Recommendation for Rejection: **The majority of reviewers (Reviewer unHu, Reviewer FWrb, Reviewer La9P, Reviewer ZTXg) expressed some concerns regarding the assumptions underlying this paper.** Suggesting inclusion of motivating examples and realistic experiments to enhance community-wide benefits from your insights.

---

> ### Author Rebuttal · Authors · 2024-08-06
>
> ### Motivational examples:
>   We agree with the suggestion to provide examples to motivate our setting. We will include these examples in the introduction on the extended version.
>
> ### Questions about the setting:
> - **1.** We focus on the model-based RL framework commonly used in theoretical works, which does not address continuous states or actions. However, our work can be extended to handle continuous states and actions by using a function approximation algorithm as a baseline and assuming the transition probabilities belong to a known parametric family.
>
> - **2.** No, there are no restrictions on the occupancy measure.
>
> ### No inconsistent notations:
> There is no inconsistency in our notations and we respectfully disagree with the reviewer on this point. We address all the specific concerns below:
>
> - **1. and 2.** In Section 4.1, we introduce the general setting of learning with expert advice, involving $K$ experts and $T$ episodes. In our algorithm, the number of experts equals the number of episodes, i.e., $K = T$. At each episode $s$ we activate an expert that remains active until the finite horizon $T$. Thus we index each expert by the episode it has been activated at.  We define the notation $[T]$ on line 21 as $[d] := \{1, \ldots, d\}$ for all $d \in \mathbb{N}$. This definition is used consistently throughout the paper. For intervals starting from an episode $t \neq 1$, we explicitly write $[t, T]$.
>
> - **3. Length $N$:** Yes, each episode consists of $N$ steps, a common assumption in episodic MDP. Practical examples include daily tasks where each day is an episode discretised within $N$  time steps.
>
> - **4. Notation $N$:** The notation $ N_{n,x,a}^{s,t}(x') $ represents the number of times an agent transitions from the state-action pair $(x, a)$ to the state $x’$ at time step $n$ between episodes $s$ and $t$. The notation $N$ indicates the length of an episode. To avoid confusion, we will change the letter for one of these notations.
>
> - **5. round $t$:** Round $t$ is the same as episode $t$.
>
> - **6. exponent symbols:**  The $2/3$ in the regret expression at Line 268 is an exponent. The $t$ in $F^t$ or $\hat{p}^t$ indexes the objective functions or probability estimation at episode $t$, as defined in Lines 115/116 and Line 118, and is used consistently throughout the paper. This is a common notation in the literature used to avoid having all indexes written as subscripts when many indexes are needed, see for example [Perrin et. al, 2020].
>
> ### Questions about MetaCURL
> - **1. and 2.** We do not fully understand the question. Could you please provide more details?
>
> - **3.** The distribution $h\_n^t$ is entirely unknown to the learner. However, since $g\_n$ is assumed to be known and each agent observes their own state-action trajectory, they can determine their external noise trajectory by simply inverting $g\_n$, that is commonly an additive or multiplicative function with respect to the noise.
>
> - **4.** There is no need to learn it. If an expert is active at episode $t$, it will output a policy; if not, it will not have an output. This is how the meta-algorithm knows which experts are active.
>
> - **5.** The runtime is determined by the number of experts (i.e., instances of the black-box algorithm) multiplied by the time complexity of the black-box algorithm. Given our choice of naive intervals for running each black-box algorithm, there are $T$ experts, which increases the computational complexity by a factor of $T$. However, as noted in Remark 3.1, there are more sophisticated ways for designing running intervals that can reduce the computational complexity to $\log(T)$ and can be adapted to our case.
>
> - **6.**
> 	- **6.1**
> 		The algorithm's error is measured by the dynamic regret in Eq. (5), which is the difference between the learner's total loss and that of any policy sequence. In environments with changing objective functions, the regret bound depends on the total variation of the objective functions or the policy sequence, $\Delta^{\pi^*}$.
>
> 		For environments with changing probability transitions, the regret bound depends on either the abrupt variation, $\Delta^p_\infty$, or the smooth variation, $\Delta^p$. A robust algorithm  depends on the minimum of these two metrics. If there are few abrupt changes, $\Delta^p\_\infty$ is smaller, and the algorithm performs at $\sqrt{\Delta^p_\infty T}$. If changes are frequent but minor, $\Delta^p$ is smaller, and the algorithm performs at $T^{2/3} (\Delta^p)^{1/3}$. Prior to our work, only [44] had an algorithm demonstrating such robustness.
>
> 		Our algorithm is the first to achieve optimal performance with respect to both $\Delta^p$ and $\Delta^p\_\infty$ and depends on $\Delta^{\pi^*}$ rather than variations in the objective function. This allows our algorithm to excel even when the objective function changes arbitrarily, a capability not offered by previous work.
>
> 	- **6.2** Our regret bound also depends on the state space $\mathcal{X}$, the action space $\mathcal{A}$, and the episode length $N$. Due to these dependencies making the regret bound expression cumbersome, we simplify the presentation by expressing the result in the order of the number of episodes $T$ alone. The notation $\tilde{O}$ signifies that the bound is of the same order as the expression in $T$, disregarding constants and logarithmic factors. Also, in Propositions 5.2, 5.3 and Appendix F we explicitly detail these dependencies. This is common practice in the literature, see for example [44].
>
> - **7.**
> 	We recognize that experiments are valuable for assessing an algorithm's practical performance, but our paper focuses on developing an algorithm that theoretically achieves the optimal regret bound within a specific framework.
>
> ### References:
> - *Perrin et. al 2020, Fictitious Play for Mean Field Games: Continuous Time Analysis and Applications, NeurIPS 2020*

---

> > ### Comment · Reviewer_FWrb · 2024-08-09
> >
> > Thank you for your detailed response. While I appreciate the clarifications provided, I remain concerned about certain aspects of your problem settings and their practical applications.
> >
> > I strongly recommend the authors to add the clarifications above to their paper.
> >
> > > we introduce the general setting of learning with expert advice, involving $K$ experts and $T$ episodes. In our algorithm, the number of experts equals the number of episodes, i.e., $K = T$. At each episode $s$ we activate an expert that remains active until the finite horizon $T$.
> >
> > This raises a key question. What if  $K \neq T$? Additionally, regarding my earlier inquiries about Learning with Expert Advice (LEA), are all your experts considered optimal? If some experts are not optimal (e.g., expert $k_1$ has a performance of 1, while expert $k_2$ has a performance of 0.9), how does your algorithm accommodate such variability?
> >
> > Furthermore, let me add more details to my previous questions. If the experts propose different actions under similar circumstances ($s_1$, expert 1 takes $a_1$ but expert 2 takes $-a_1$), how does your algorithm resolve these conflicts? Addressing these scenarios would significantly enhance the robustness and applicability of your approach.
> >
> > The authors mention that "this problem can be reduced to the sleeping expert problem." Could you provide further explanation on this point? Specifically: In L218,  The authors mention that "experts are not required to provide solutions at every time step". What if, at a certain step, no experts provide solutions? Is there a minimum number of experts that must be "awake" or active for the algorithm to function effectively?
> >
> > Could you clarify what do you mean by "that remains active until the finite horizon $T$"?
> >
> > > $g_n$ is assumed to be known.
> >
> > Is this assumption necessary? What if $g_n$ is not known? How could we validate it in real-world scenarios?
> >
> > > The notation $N(X')$ represents the number of times an agent transitions from the state-action pair. The notation $N$ indicates the length of an episode. To avoid confusion, we will change the letter for one of these notations.
> >
> > I appreciate your willingness to revise the notation to reduce confusion. This supports my initial observation that your current notation is not very consistent. As such, I strongly recommend including a notation table in the paper to clearly define and distinguish all critical symbols and terms. This would greatly aid in understanding and improve the readability of your work.
> >
> > > We recognize that experiments are valuable for assessing an algorithm's practical performance, but our paper focuses on developing an algorithm that theoretically achieves the optimal regret bound within a specific framework.
> >
> > Providing numerical examples or visual figures that illustrate real-world applications would greatly strengthen your paper. Such examples would address concerns raised by other reviewers, including Reviewer unHu, regarding the practicality and applicability of your algorithm. Empirical evidence, even in simplified scenarios, would offer a more comprehensive evaluation of your proposed method. E.g., could you provide an intuitive numerical example based on your global rebuttal (energy grid optimization, or robotics or finance)?
> >
> > Additionally, in your newly added reference, [Perrin et. al, 2020] have some experiments, and figures clearly showing their results.
> >
> > > Our regret bound also depends on the state space, the action space, and the episode length.
> >
> > As such, how your algorithm will perform, when their dimensions are increasing?

---

> > > ### Author Response · Authors · 2024-08-09
> > >
> > > Thank you for your quick response. We believe there might unfortunately be some misunderstandings. Below, we offer our point-by-point responses to address and clarify these concerns.
> > >
> > > > This raises a key question. What if $K \neq T$?
> > >
> > > **Answer:** This scenario is not possible because the core principle of our meta-algorithm (Alg. 1) is specifically to design $K=T$ experts. This approach is a well-established technique in several dynamic online learning studies (see [13, 22, 24] cited in the paper). Additionally, as stated in lines 225-226, an expert is simply an instance of the black-box algorithm launched by the meta-algorithm. Therefore, we can have as many experts as necessary (the number is controlled by the meta-algorithm).
> > >
> > > > […] are all your experts considered optimal? If some experts are not optimal (e.g., expert $k_1$ has a performance of 1, while expert $k_2$ has a performance of 0.9), how does your algorithm accommodate such variability?
> > >
> > > **Answer:**  We don't quite understand your point. Naturally, not all experts are optimal. The purpose of the meta-algorithm is to perform as well as the best expert, which aligns with the standard regret objective in Learning with Expert Advice. Regardless of the experts' performance, our theoretical guarantee in Thm. 5.1 remains valid.
> > >
> > > >  If the experts propose different actions under similar circumstances ($s_1$, expert 1 takes $a_1$ but expert 2 takes $-a_1$), how does your algorithm resolve these conflicts?
> > >
> > > **Answer:** Experts proposing different actions under similar circumstances is not a problem. As is standard in the literature on learning with expert advice (see *Prediction, Learning, and Games* by Cesa-Bianchi and Lugosi, 2006), the experts' advice—represented as different state-action distributions—are combined in line 7 of Alg. 1. The resulting combined state-action distribution is then used to select the action to be taken.
> > >
> > >
> > > > "this problem can be reduced to the sleeping expert problem." Could you provide further explanation on this point? Specifically: In L218, The authors mention that "experts are not required to provide solutions at every time step". What if, at a certain step, no experts provide solutions? Is there a minimum number of experts that must be "awake" or active for the algorithm to function effectively?
> > >
> > > **Answer:** This will never be the case. An expert is simply an instance of the black-box algorithm. By design, we initialize one black-box algorithm at the beginning of each episode, which continues to provide outputs throughout all episodes until $T$. Thus, in every episode, there is always an expert providing solutions. The expert corresponding to the black-box instance initialized in episode $t$ is not active in any earlier episode $t’ < t$. However, all experts initialized in episodes $s < t’$  are active at episode $t$.
> > >
> > >
> > > > $g_n$ known. Is this assumption necessary? What if $g_n$ is not known? How could we validate it in real-world scenarios?
> > >
> > > **Answer:** Yes, this assumption is necessary in our framework. We explain the importance of addressing CURL and this assumption in our general response to all reviewers. If $g_n$ is unknown but belongs to a known family of parametric functions, we could extend our analysis. However, if $g_n$ is entirely unknown, it would be more challenging and would require further work, as already mentioned in the Conclusion (line 319) of our paper. We've already provided several examples of real-world scenarios where the dynamics can be modeled under the assumption of $g_n$ being known, with an unknown and non-stationary external noise distribution $h_n^t$ (see lines 142-154 in the paper and our general response).
> > >
> > > > This supports my initial observation that your current notation is not very consistent. As such, I strongly recommend including a notation table in the paper to clearly define and distinguish all critical symbols and terms. This would greatly aid in understanding and improve the readability of your work.
> > >
> > > **Answer:** The notation is not inconsistent. The function N can easily be distinguished from index N both by its signature and the presence of indices. We find the statement that: "since we accept to modify this notation, it supports the claim of inconsistency of all our notations" to be unjustified. The purpose of the review process is to improve the paper, and using our willingness to make changes to satisfy the reviewer as an argument against the paper is non-constructive and unfair. Nonetheless, we can easily implement this modification in the final version, which we see as a minor issue, and believe it should not be used as a reason to reject the paper. We will include a notation table in the final version.

---

> > > > ### Comment · Reviewer_FWrb · 2024-08-09
> > > >
> > > > **Empirical validation**: While it is acceptable for a NeurIPS paper to focus on theoretical contributions without extensive experiments, I share similar concerns with Reviewer 55Y9. It would significantly strengthen the paper if the proposed framework were tested on some empirical baselines. This would not only validate the theoretical claims but also demonstrate their practical relevance.
> > > >
> > > > Furthermore, as Reviewer ZTXg pointed out, the dynamic hypothesis presented in L128 seems to be a strong assumption. This assumption necessitates empirical evidence to show that it holds in practicem e.g., in infinite horizon or continuous spaces. Although the authors mentioned some real-world applications in their rebuttal, I remain unconvinced. It is particularly important to clarify how your assumptions (e.g., $K=T$, assumptions about experts, etc. ) align with real-world scenarios, such as those in energy grid optimization, robotics, or finance.
> > > >
> > > > Furthermore, as an average reader of RL literature, I still found the notations in the paper challenging to follow. Including a notation table and additional clarifications would greatly enhance the readability and accessibility of your work. I still find some of the assumptions in the paper to be unconvincing. The motivation behind these assumptions is not entirely clear, especially in the absence of a motivating example. It is a little difficult to see how the broader RL community could readily apply or benefit from the theoretical insights offered in this paper.
> > > >
> > > > Given these concerns, I am unable to raise my score at this time and will maintain my rating of 4.

---

> > > > > ### Author Response · Authors · 2024-08-10
> > > > >
> > > > > We address below the reviewer comments:
> > > > >
> > > > > > Empirical validation: While it is acceptable for a NeurIPS paper to focus on theoretical contributions without extensive experiments, I share similar concerns with Reviewer 55Y9. It would significantly strengthen the paper if the proposed framework were tested on some empirical baselines. This would not only validate the theoretical claims but also demonstrate their practical relevance.
> > > > >
> > > > > **Answer:** We respectfully disagree with the reviewer's perspective and believe that the absence of empirical evaluations in a theoretical paper should not be a reason for rejection. Our paper contributes significantly to the theoretical understanding of non-stationary CURL, an area with no prior research. It's important to highlight that many foundational papers on non-stationary RL were published without experiments [38, 25, 15, 40, 27, 12, 11, 36, 16, 44]. We regret that the reviewer may not fully recognize the value of our contributions.
> > > > >
> > > > >
> > > > > > Furthermore, as Reviewer ZTXg pointed out, the dynamic hypothesis presented in L128 seems to be a strong assumption. This assumption necessitates empirical evidence to show that it holds in practicem e.g., in infinite horizon or continuous spaces. Although the authors mentioned some real-world applications in their rebuttal, I remain unconvinced. It is particularly important to clarify how your assumptions (e.g., $K=T$, assumptions about experts, etc. ) align with real-world scenarios, such as those in energy grid optimization, robotics, or finance.
> > > > >
> > > > > **Answer:** The dynamic assumption may appear strong, but it is valid in practical situations. For instance, [34] provides a real-world motivating application that aligns perfectly with our assumption and framework, addressing a significant problem in the context of climate change. We also believe that our algorithm could be extended in a future work to deal with continuous spaces using function approximation approaches. We point out that this extension has nothing to do with the dynamic’s assumption.
> > > > >
> > > > > Moreover, although the reviewer may not fully appreciate it, the CURL framework is significantly more challenging than the classical RL framework, where the loss is linear. Managing convex loss functions under partial feedback is considerably more complex than dealing with linear ones (*e.g.*, no efficient optimal algorithm exists for convex bandits while the problem is mostly solved for multi-armed bandits), and it necessitates certain assumptions. No existing work on CURL offers theoretical regret guarantees without this dynamic assumption.
> > > > >
> > > > > The reviewer's concerns about assumptions, such as $K = T$ and expert assumptions, indicate a misunderstanding of how prediction with expert advice algorithms are used as meta-procedures. These are not assumptions but rather the design of meta-algorithms. Meta-algorithms are extensively used in the online learning literature to develop adaptive algorithms, as demonstrated in works such as [13, 22, 24], among others, or
> > > > >
> > > > > - *Adaptivity and Non-stationarity: Problem-dependent Dynamic Regret for Online Convex Optimization. Peng Zhao, Yu-Jie Zhang, Lijun Zhang, Zhi-Hua Zhou, 2024*
> > > > > - *Metagrad: Multiple learning rates in online learning. T Van Erven, WM Koolen - Advances in Neural Information Processing Systems, 2016*
> > > > > - *Locally-Adaptive Nonparametric Online Learning. Ilja Kuzborskij, Nicolò Cesa-Bianchi, 2020*
> > > > >
> > > > >
> > > > > > Furthermore, as an average reader of RL literature, I still found the notations in the paper challenging to follow. Including a notation table and additional clarifications would greatly enhance the readability and accessibility of your work. I still find some of the assumptions in the paper to be unconvincing. The motivation behind these assumptions is not entirely clear, especially in the absence of a motivating example. It is a little difficult to see how the broader RL community could readily apply or benefit from the theoretical insights offered in this paper.
> > > > >
> > > > > **Answer:** Including a notation table is a good suggestion that can easily be added in the paper and it should not be a reason for rejection. While our notations may seem complex, they are necessary due to the episodic online CURL framework, which involves multiple indices ($t$ for episodes, $n$ for time steps, $x$ for states, $a$ for actions, etc.). We are open to any constructive suggestions from the reviewer to improve our notations.

---

> > > ### Author Response · Authors · 2024-08-09
> > >
> > > We continue below to address each point of the reviewer.
> > >
> > > > Providing numerical examples or visual figures that illustrate real-world applications would greatly strengthen your paper. […] in your newly added reference, [Perrin et. al, 2020] have some experiments, and figures clearly showing their results.
> > >
> > > **Answer:** We emphasize that this is a paper on theoretical reinforcement learning and online learning. The work by Perrin et al. (2020) addresses an offline scenario in mean field reinforcement learning with fully known dynamics (both $g_n$ and $h_n$ are known) and assumes stationary dynamics and losses, making it relatively straightforward to design experiments for such a setting.
> > >
> > > In contrast, our framework is considerably more complex, involving adversarial losses and changing dynamics. Conducting experiments for adversarial and non-stationary MDPs is particularly challenging due to the difficulty of constructing optimal policies across episodes. The existing literature largely lacks experimental validation for such scenarios. Most of the influential papers addressing non-stationarity in reinforcement learning that we cite do not include experiments, such as [38, 25, 15, 40, 27, 12, 11, 36, 16, 44], among others.
> > >
> > > > As such, how your algorithm will perform, when their dimensions are increasing?
> > >
> > > **Answer:** As previously mentioned, this dependency in each term of the regret is explicitly outlined in Propositions 5.2, 5.3, 5.4 and Appendix F. The final dependency in Theorem 5.1 is influenced by the choice of the black-box algorithm. For the Greedy MD-CURL algorithm we propose, the dependency on the constants is the same as in [38]: $N^2 \mathcal{X} \sqrt{\mathcal{A}}$, multiplied by the dependency on $T$ and the variation budgets as has been outlined in Appendix G.

---

> ### Comment · Reviewer_FWrb · 2024-08-12
> **Recommendation for Rejection: Suggesting Inclusion of Motivating Examples and Realistic Experiments to Enhance Community-Wide Benefits from Your Insights**
>
> Still, from my perspective, a theoretical paper should only be accepted at NeurIPS if it clearly demonstrates its motivation, and its assumptions are valid in practice (as mentioned by Reviewer unHu, Reviewer FWrb, Reviewer La9P, Reviewer ZTXg); and the whole RL community can benefit from its insights.
>
> As the authors mention Robotics and Finance in their global response, here are some open repositories I recommend they test:
>
> [1] https://github.com/openai/gym
>
> [2] https://github.com/google-deepmind/dm_control
>
> [3] https://github.com/AI4Finance-Foundation/FinRL
>
> For example, many existing control and robotics problems involve infinite horizons and continuous actions and states. How can we assume a large number of experts are possible? How can your assumptions make sense in such scenarios? Additionally, in finance, many high-frequency trading problems occur within milliseconds, making it impractical for all your assumptions to be true.
>
> > Moreover, although the reviewer may not fully appreciate it, the CURL framework is significantly more challenging than the classical RL framework, where the loss is linear.
>
> I agree CURL is more challenging. As such, it is more important to clearly on experiments show the proposed method is able to outperform the existing methods.
>
> > This scenario is not possible because the core principle of our meta-algorithm (Alg. 1) is specifically to design $K=T$ experts. Therefore, we can have as many experts as necessary (the number is controlled by the meta-algorithm).
>
> Why is such a scenario considered impossible in your settings? In practice, it's common to encounter extremely long horizons or a high number of episodes, yet with a limited number of experts. Still, your settings appear overly restrictive.
>
> > Including a notation table is a good suggestion that can easily be added to the paper and should not be a reason for rejection.
>
> Actually, I need to ensure your theory is correct. While reading, I still find lots of notations should be better and properly defined. Without clear notation, it is extremely difficult for me to evaluate it. Therefore, I still have some concerns regarding your theoretical contributions.
>
> > The dynamic assumption may appear strong, but it is valid in practical situations. For instance, [34] provides a real-world motivating application that aligns perfectly with our assumption and framework, addressing a significant problem in the context of climate change. We also believe that our algorithm could be extended in a future work to deal with continuous spaces using function approximation approaches. We point out that this extension has nothing to do with the dynamic’s assumption.
>
> I strongly recommend that the authors directly test their framework on such benchmarks, instead of merely discussing it.
> This claim remains questionable.

---

### Official Review · Reviewer_55Y9 · 2024-07-13

**Soundness:** 3
**Presentation:** 3
**Contribution:** 3
**Rating:** 6
**Confidence:** 2

**Summary:**

The paper studied the Concave Utility Reinforcement Learning (CURL) problem in non-stationary environments, which extends classical reinforcement learning (RL) to handle convex performance criteria in state-action distributions induced by agent policies. The paper proposed MetaCURL to address the challenges posed by non-stationary Markov Decision Processes (MDPs) with changing losses and probability transitions by using a meta-algorithm that aggregates multiple black-box algorithm instances over different intervals. The algorithm achieves optimal dynamic regret without requiring prior knowledge of the MDP changes, making it suitable for adversarial losses and providing a significant advancement in the RL community.

**Strengths:**

1. The designed algorithm is parameter-free, and does not require the knowledge of variation budget of transitions.
2. The results are theoretically solid.

**Weaknesses:**

1. The model falls into the tabular MDPs, where the state-action pairs are finite. I am wondering if similar technique can be applied to the case of function approximation. In addition, there is a missing related work on the non-stationary RL with function approximation [1].
2. The transition dynamics, while there are many suitable applications, is also restrictive for a theoretical work.


Feng et al., Non-stationary Reinforcement Learning under General Function Approximation, ICML, 2023.

**Questions:**

Please see the weaknesses part.

**Limitations:**

No further limitations to be addressed

---

> ### Author Rebuttal · Authors · 2024-08-06
>
> We thank the reviewer for their comments and for recognizing the new theoretical insights in our paper. We address the questions below:
>
> - **1. tabular MDP:** Thank you for bringing this related work to our attention, we will include a citation in our paper. Thanks for raising this question, we believe this could be an interesting future work of our paper and we will add it to the conclusions. To answer it, we can break the necessary changes to adapt our result to function approximation in two:
>
> 	- **The Meta algorithm:** the analysis of the meta-algorithm depends on the method used to estimate $ p^t $. We believe that a parametric estimation could be employed, which would require a new loss function for the second expert algorithm that handles non-stationarity on the estimation of the probability transitions.
>
> 	- **The Black-Box Algorithm:** We need to select a black-box algorithm for CURL that generalizes well to function approximation scenarios. Just as policy optimization algorithms are known to generalize effectively in such scenarios for RL (e.g., see [Luo et al. 2021]), we believe that the black-box approach we propose from [35] in Appendix G, which adapts policy optimization for CURL, can be extended to this context.
>
> - **2. Transition dynamics:** We address this concern in our general response to all reviewers as well. We open on some details below:
> 	- **New theoretical analysis:** Addressing the non-stationary CURL scenario is theoretically more challenging than RL, even with restrictive assumptions about the dynamics. To derive a policy using the learning with experts advice framework, we must estimate the losses of non-played expert policies based solely on the observations from the played policy.
>
>          If $ g\_n $ is known, we can use the external noises observed by the agent and the function $g\_n $ to simulate the trajectory of any non-played policy in the true environment, thereby defining an empirical state-action distribution $\hat{\mu}$ for each expert. In RL, where $ F^t(\mu) := \langle \ell^t, \mu \rangle $, estimating the loss of each expert at episode $ t $ by taking $\langle \ell^t, \hat{\mu} \rangle$ is sufficient to prove meta-regret convergence due to the linearity of the expression, requiring only common model-based RL techniques.
>
>         In CURL, this approach fails due to its non-linearity convexity. Therefore, we need to address the more challenging task of estimating the non-stationary probability transition $ p^t $ to estimate the losses. This leads to the main theoretical innovation of the paper: the development of the second expert scheme capable of estimating the transition probabilities even in the presence of non-stationarity. The non-linearity of CURL makes it theoretically more challenging than RL.
>
> 	- **Existence of black-box algorithm for CURL that explores:** No closed-form algorithm for exploration in CURL exists. Policy optimization (PO) methods for RL rely on the value function, making them unsuitable for CURL, as CURL invalidates the classical Bellman equations. The alternative occupancy-measure methods, while applicable, are computationally less efficient and lack a closed-form solution. The algorithm we propose as a black-box is a PO method adapted for CURL and is derived from [35], but it also assumes that $ g_n $ is known.
>
> ### References:
> - *Luo et al. 2021, Policy Optimization in Adversarial MDPs: Improved Exploration via Dilated Bonuses, NeurIPS 2021*

---

> > ### Comment · Reviewer_55Y9 · 2024-08-13
> >
> > Thanks for the response. I decide to maintain my score.

---

### Official Review · Reviewer_La9P · 2024-07-14

**Soundness:** 3
**Presentation:** 2
**Contribution:** 2
**Rating:** 5
**Confidence:** 2

**Summary:**

The authors present a policy learning algorithm for non-stationary (+ uncertain) environments and convex utilities.
The proposed algorithm is a meta-algorithm which runs multiple black-box algorithms and aggregates outputs with something they call a sleeping expert framework. The algorithm achieves optimal dynamic regret.

**Strengths:**

- Proposed an algorithm for non-stationary and uncertain convex utility MDP
- Achieves optimal dynamic regret

**Weaknesses:**

- The paper considers uncertain/non-stationary noise distribution. However, until 1/3rd of the paper, I got an impression of uncertain dynamics (without any structure assumption).
- The paper is difficult to follow particularly section 3 from line 175 and section 4

**Questions:**

- Are baselines the same as the black-box algorithms?
- Line 164, what does parametric algorithm mean in this context
- Line 141 why do policy optimization algorithms have lower complexity?
- Any thoughts on how the algorithm changes if g_n is uncertain?

Comments for improving paper:
- line 13: we achieve ---> the algorithm achieves?
- line 15: "full adversarial losses, not just stochastic ones." is ill-posed
- I am familiar with both non-stationarity and uncertainty. However, since the paper builds on these two challenges, I would suggest explaining them (at least in a high-level) early in the introduction. Similarly, for the terms adversarial objectives and learner (line 50).
- Line 133-135: Please include it in the introduction (could be a bit vague) but it's important to note that the structure of MDP is already known and only the noise is uncertain.
- Maybe write contributions as a paragraph, since points 2,3 are not contributions but explanations of point 1.

---

> ### Author Rebuttal · Authors · 2024-08-06
>
> We thank the reviewer for their comments and questions. We agree that the paper is notation-heavy and may be difficult to follow in some parts. We welcome any further suggestions on how to improve the readability of the paper.
>
> ### Questions
> - **Baselines vs blackbox:** Yes, baselines algorithms are the same as blackbox, we apologize for the confusion caused by using both terms. We will revise this to ensure consistency.
>
> - **Line 164** We apologize for the typo where we incorrectly used the word "parametric." We wanted to say that our approach is applicable to any black-box algorithm that satisfies Equation (10), and specifically it also works if the algorithm depends on a learning rate $\lambda$, without requiring us to specify the optimal $\lambda$ as an input (parameter-free).
>
> - **Line 141** In tabular RL, there are two main approaches for dealing with uncertainty and adversarial losses: policy optimization (PO) and occupancy-measure (or state-action distribution) methods.
>
> 	Occupancy-measure methods leverage the ideas from UCRL2 [4]. They construct a set of plausible MDPs compatible with the observed samples and then play the policy that minimizes the cost over this set of plausible MDPs.
>
> 	 PO methods evaluates the value function and uses a mirror descent-like updates directly on the policy, solving the optimization problem through dynamic programming to obtain a closed-form solution for the policy. This approach is more computationally efficient than planning in the space of all plausible MDPs.
>
> 	For a more detailed discussion on both approaches we refer to the works of [Luo et al. 2021, Efroni et al. 2020a,b].
>
> - **How the algorithm changes with $g\_n$ unknown:**
> 	- **MetaCURL analysis:** MetaCURL with a black-box algorithm that explores could be extended to the case where $g_n$ is unknown but belongs to a known parametric family. The  case where $g_n$ is completely unknown would be harder and require further work.
> 	- **Existence of black-box algorithm for CURL that explores:** No closed-form algorithm for exploration in CURL exists. PO methods for RL rely on the value function, making them unsuitable for CURL, as CURL invalidates the classical Bellman equations. The alternative occupancy-measure methods, while applicable, are computationally less efficient and lack a closed-form solution. The algorithm we propose as a black-box is a PO method adapted for CURL and is derived from [35], but it also assumes that $ g_n $ is known.
>
> ### Comments for improving the paper:
> Thank you for pointing out the typos in lines 13 and 15; we will correct them. We also appreciate the feedback on making our paper more accessible to readers outside the online learning and theoretical reinforcement learning communities. We will provide a clearer explanation of the challenges of non-stationarity, uncertainty, and adversarial losses earlier in the introduction.
>
> We will clarify our assumptions on the dynamics by moving Equation (8) to the introduction. We just want to highlight that we fully acknowledge the limitations of our restrictive assumptions in the paper. These are addressed in the abstract, introduction, comparison table 1, Section 2, and the conclusion, where we also point it as an important improvement for future work.
>
> Regarding our contributions, we list them as separate points to highlight the distinct novelties of our method. The third point, in particular, stands as an independent contribution:
> - **Point $1$, New algorithm:** MetaCURL is a new-algorithm for online MDPs dealing with non-stationarity, partial uncertainty, adversarial losses, convex losses, achieving optimal dynamic regret, and being efficient to compute.
> - **Point $2$, New analysis:** this is the first application of learning with expert advice theory to Markov Decision Processes, resulting in analyses significantly distinct from existing RL approaches and potentially inspiring new algorithms for RL and CURL.
> - **Point $3$, New black-box algorithm:** Developing the meta algorithm and proving its regret is one aspect (points $1$ and $2$), while demonstrating the existence of a baseline algorithm that meets the regret bound in Equation (10) is another contribution, detailed in Section 5 and Appendix G.
>
> ### References:
> - *Luo et al. 2021, Policy Optimization in Adversarial MDPs: Improved Exploration via Dilated Bonuses, NeurIPS 2021*
> - *Efroni et al. 2020a, Optimistic Policy Optimization with Bandit Feedback, ICML 2020*
> - *Efroni et al. 2020b, Exploration-Exploitation in Constrained MDPs*

---

> ### Comment · Reviewer_La9P · 2024-08-10
>
> I thank the authors for their replies. I went over them and do not have any further questions. Please include the changes as promised in your revised version, particularly regarding the dynamics assumption and improving readability.

---

### Official Review · Reviewer_kKtv · 2024-07-15

**Soundness:** 3
**Presentation:** 3
**Contribution:** 3
**Rating:** 7
**Confidence:** 1

**Summary:**

This paper presents theoretical results on the CURL algorithm for non-stationary MDPs. The proposed meta CURL models non-stationarity factors using external noise and achieves low dynamic regret in near-stationary environments, with regret only related to the frequency and magnitude of changes. Overall, the work provides solid theoretical results for CURL and non-stationary RL. Although I am not an expert on RL theory (my focus is more on algorithms and applications), I would give an acceptance rating for this initial review and will be engaged in the discussion.

**Strengths:**

- [**Motivation and Significance**]: The problem of non-stationary MDPs is critical in CURL, and this paper provides valuable theoretical insights and guarantees for the CURL algorithm. While I am not an expert in RL theory, I think in general this work is critical for both RL theory and empirical algorithms design.
- [**Technical Soundness**]: While I am not an expert in RL theory, the theoretical results and proofs presented in the paper appear to be solid and well-constructed, especially on the regret analysis part (only related to changing frequency and magnitude).
- [**Presentation**]: The paper is generally well-written and the theoretical analysis is clearly presented (accessible even to readers outside the theory domain)

**Weaknesses:**

Since I am not an expert on RL theory, I have listed most of my questions in this section.

Q1: In equation (8), the noise captures the non-stationary factors. Will these factors from different parts of the MDP (such as the reward function or state dynamics) separately influence or impact the major theory and regret bound?

Q2: In some non-stationary or meta RL works, the non-stationary or distribution change factors are not only from external noise but also from the agent’s policy itself. How does this affect the major theoretical results? Any comments on this point would be appreciated.

**Questions:**

I put the questions in the above section.

**Limitations:**

I do not think this theoretical work could pose any potential negative societal impact.

---

> ### Author Rebuttal · Authors · 2024-08-06
>
> We thank the reviewer for their comments and for recognizing the new theoretical insights in our paper. We address the questions below:
>
> - **Q1:** We take into account two types of non-stationarity: on the objective functions and on the dynamics from the external noise distribution $h_n^t$. Each impact the regret in a different way in Theorem 5.1:
> 	- **Non-stationarity factors from the objective functions:** Our regret bound works for adversarial objective functions, i.e. that can change arbitrarily at each episode, and are unknown to the learner. This robustness to non-stationarity in losses affects the regret through the term $\Delta^{\pi^*}$. This is a novel aspect of our algorithm. We are the first to present an algorithm that addresses both dynamic non-stationarity and arbitrarily changing losses without requiring prior knowledge of the variation budget.
> 	- **Non-stationarity in dynamics:** In the paper, non-stationarity in the dynamics arises from the unknown distribution of the external noises $ h\_n^t $. This affects the regret through the terms $\Delta^p$ and $\Delta^p\_\infty$, as detailed in Theorem 5.1. However, our bounds would still hold if the state dynamics were also non-stationary (i.e., $ g\_n^t $ in line 128 instead of just $ g\_n $), provided $ g\_n^t $ is known to the learner.
>
> - **Q2:** We also account for the non-stationarity of the policies in our work through the term $\Delta^{\pi^{\*}}$, defined in Equation (7). Our algorithm achieves the optimal regret bound for this term as stated in Theorem 5.1. Could you please provide the references to the works you mentioned so that we can offer more details on this?

---

> > ### Comment · Reviewer_kKtv · 2024-08-10
> >
> > Thank you for your detailed response. My concerns have been mostly addressed. As to the Q2, one reference could be [1] where the non-stationary factors could come from different sources in MDPs.
> >
> > [1] Xie, Annie, James Harrison, and Chelsea Finn. "Deep reinforcement learning amidst continual structured non-stationarity." International Conference on Machine Learning. PMLR, 2021.

---

> > > ### Author Response · Authors · 2024-08-12
> > >
> > > Thank you for your response, we are glad to have addressed most of the reviewer concerns. We hope to address  Q2 below, based on reference [1].
> > >
> > > In [1], non-stationarity is represented by a probabilistic model. Indeed, the Markov decision process (including both the rewards and the probability transition kernels) is indexed by some latent variable $z$ that evolves according to a hidden Markov model. Our paper does not assume any model for how the non-stationarity arises, making it a robust approach.
> > >
> > > From the perspective of losses, our algorithm is robust to adversarial losses—losses that can change arbitrarily and are unknown to the learner—affecting our regret bound through  $\Delta^{\pi^*}$.
> > >
> > > From the perspective of the probability transition kernel, we do not make any model assumptions about the distribution of the non-stationarity factors, $h_n^t$. The only assumption is that $g_n$ from Equation (8) is known. However, we believe our approach could be extended to the case where $g_n$ belongs to a known family of parametric functions with parameters that are unknown and vary across episodes, also without assuming any specific model for the distribution of these parameter changes.

---

### Official Review · Reviewer_ZTXg · 2024-07-30

**Soundness:** 3
**Presentation:** 2
**Contribution:** 3
**Rating:** 7
**Confidence:** 3

**Summary:**

This paper addresses online learning in non-stationary episodic loop-free Markov decision processes (MDPs) with changing losses and probability transitions. It extends the Concave Utility Reinforcement Learning (CURL) problem to handle convex performance criteria in state-action distributions, overcoming the non-linearity that invalidates traditional Bellman equations. The introduced MetaCURL algorithm, the first for non-stationary MDPs, runs multiple black-box algorithm instances over different intervals, using a sleeping expert framework to aggregate outputs. MetaCURL achieves optimal dynamic regret under partial information without prior knowledge of MDP changes, handling fully adversarial losses. This approach is expected to be of significant interest to the RL community.

**Strengths:**

The novelty is clear. That is expanding CURL in non-stationary MDPs.

**Weaknesses:**

My comments on weaknesses are listed below.

**Questions:**

Thanks for the opportunity to review this paper. I have some questions as follows.
[Major issues]

1. Novelty

- The statement "non-requiring prior knowledge of the environment's variations" (lines 68-69) needs clarification. In Table 1, the dynamic regret of the proposed algorithms still depends on $\Delta^p_\infty$ and $\Delta^p$, which seem to contain information about environmental changes (lines 123-126). How does this align with the claim of not requiring prior knowledge?

2. Dynamic regret representation

- The use of $\Delta^{\pi^\star}_t$ to represent the upper bound of dynamic regret requires further justification. While it may seem reasonable since the optimal policy is defined given the environment, what happens if the optimal policy is not unique? In such cases, $\Delta^{\pi^\star}_t$ could take multiple values, as there can be multiple optimal policies at times $t$ and $t+1$.

3. Dynamic hypothesis

- The dynamic hypothesis presented in line 128 appears to be a strong assumption. Reinforcement learning (RL) is inherently about learning from data, and assuming prior knowledge of how the environment operates seems misaligned with RL's principles. This assumption aligns more with traditional control theory, where the plant dynamics are known. In RL, even in model-based RL, the model is estimated, and the policy is learned simultaneously.

[Minnor issues]
1. Terminology
- The term "loop-free" used in the abstract needs clarification. This term is not commonly used in the context of learning but is more prevalent in control theory. Is "loop-free learning" akin to continual learning with infinite episodes? More precise terminology would help readers understand your contributions better.

2. Regarding with the related works.
- It seems the paper lacks some recent related works (around 2023,2204)  in theoretical non-stationary RL works. Please include the following to keep the track.

 [1] Lee, H., Ding, Y., Lee, J., Jin, M., Lavaei, J., & Sojoudi, S. (2024). Tempo adaptation in non-stationary reinforcement learning. Advances in Neural Information Processing Systems, 36.
[2] Feng, S., Yin, M., Huang, R., Wang, Y. X., Yang, J., & Liang, Y. (2023, July). Non-stationary reinforcement learning under general function approximation. In International Conference on Machine Learning (pp. 9976-10007). PMLR.

**Limitations:**

There is no limitations on potential negative societal impact .

---

> ### Author Rebuttal · Authors · 2024-08-06
>
> ### Questions
> - **Novelty:** Our algorithm runs without needing knowledge of the environment changes as input, but the final error guarantee does depend on these variations, specifically through the non-stationarity measures $\Delta^p$ and $\Delta^p_\infty$. This agrees with the lower bound in [33], which we cite in line 68 of the paper. This lower bound is independent of the algorithm and captures the difficulty of the problem as a function of these measures. Note that all previous algorithms in the literature but the one in [44] require knowledge of the environment changes as input, which in practice is often not feasible. This is an advantage of our algorithm.
>
> - **Dynamic regret representation:** Here, $(\pi^{\*,t})\_{t \in [T]}$ refers to any sequence of policies, not just the optimal ones. The dynamic regret in Equation (5) and the non-stationarity measure $\Delta\_t^{\pi^{*}}$ in Equation (7) are defined for any sequence of policies $(\pi^{\*,t})\_{t \in [T]}$. Therefore, our regret bound is valid for any sequence of policies. Choosing the optimal sequence of policies to evaluate the regret is natural. The fact that this sequence of policies may not be unique poses no problem in which case we would pay $\min\_{\pi^{\*}} \Delta^{\pi^{\*}}$.
>
>
> - **Dynamic Hypothesis:** We further discuss the importance of the dynamics assumption in Eq. (8) in the common response to all reviewers. We want to emphasize that our dynamics still assume that the noise's distribution $h_n^t$ and the cost functions are completely unknown, requiring simultaneous model learning and optimization. This places our problem outside the scope of control literature and into the realm of reinforcement learning. Many existing RL studies address problems other than the exploration-exploitation dilemma, such as the initial work on concave utility RL mentioned in the paper, which typically consider offline settings with fully known dynamics [23, 47, 48, 5, 46, 20].
>
> - **Terminology:** We appreciate the reviewer's comment and agree that the term may be ambiguous. We will provide additional explanations in the paper to clarify its meaning. However, "loop-free" is also commonly used in the context of learning in model-based episodic Markov Decision Processes (MDP) to describe online MDP problems with fixed episode lengths and a fixed initial state-action distribution per episode. This term was first introduced by [Neu et al. 2010], and has since been used in several influential works, such as: [Neu et al. 2012, Zimin et al. 2013, Rosenberg et al. 2019, Jin et al. 2020, Efroni et al. 2020, Moreno et al. 2024]; among many others.
>
> - **Related work:** We greatly appreciate the reviewer for pointing us to these related works. We will include them in our citations.
>
> ### References:
> - *Neu et al. 2010, Online Markov Decision Processes under Bandit Feedback, NeurIPS*
> - *Neu et al. 2012, The adversarial stochastic shortest path problem with unknown transition probabilities, AISTATS*
> - *Zimin et al. 2013, Online Learning in Episodic Markovian Decision Processes by Relative Entropy Policy Search, NeurIPS*
> - *Rosenberg et al. 2019, Online Convex Optimization in Adversarial Markov Decision Processes, ICML*
> - *Jin et al. 2020, Learning Adversarial Markov Decision Processes with Bandit Feedback and Unknown Transition, ICML*
> - *Efroni et al. 2020, Optimistic Policy Optimization with Bandit Feedback, ICML*
> - *Moreno et al. 2024, Efficient Model-Based Concave Utility Reinforcement Learning through Greedy Mirror Descent, AISTATS*

---

> > ### Comment · Reviewer_ZTXg · 2024-08-09
> >
> > Thank you for the update.
> >
> > The reviewer still has a few questions and requires some clarification.
> >
> > - **Novelty**: The reviewer acknowledges that the upper bound should be dependent on the environment's changes. However, the statement "Our algorithm runs without needing knowledge of the environment changes as input" seems to imply a rather strong assumption. Does this mean that the algorithm requires *exact* information about the environment changes, or is it based on *estimated* information? If it requires exact information, the algorithm might lose its practicality, as it is generally impossible to know the environment's changes with precision in a time-varying setting. If this is the case, could the authors propose an upper bound that accounts for estimation error? I also recommend referring to recent non-stationary reinforcement learning papers [1,2] that incorporate estimation errors in both model-free and model-based approaches.
> >
> >   [1] Lee, H., Ding, Y., Lee, J., Jin, M., Lavaei, J., & Sojoudi, S. (2024). Tempo adaptation in non-stationary reinforcement learning. Advances in Neural Information Processing Systems, 36. – This work includes estimation error in a model-based approach.
> >
> >   [2] Lee, H., Jin, M., Lavaei, J., & Sojoudi, S. (2024). Pausing Policy Learning in Non-stationary Reinforcement Learning. In Forty-first International Conference on Machine Learning. – This work incorporates estimation error in a model-free manner.
> >
> > - **Dynamic Regret Representation**: While the author's response wasn't perfectly aligned with the reviewer's initial query, the clarification provided has fully addressed the concern. The reviewer is now persuaded by the approach of taking $ \min_{\pi^*} \Delta^{\pi^*} $ for the constant upper bound. Thank you.
> >
> > - **Dynamic Hypothesis**: The reviewer still finds the dynamic hypothesis to be a somewhat strong assumption, even though the system includes noise. This is because noise is an inevitable consideration in any system design. However, based on the prior works referenced, this issue is considered fully addressed. Thank you.
> >
> > - **Terminology**: Thank you for the clarification. Could the authors please be more specific about the term "loop-free"? What exactly does "loop" refer to, and why is the proposed algorithm described as "loop-free"?
> >
> > - **Related Work**: Thank you for the consideration. Please include these references if the paper is accepted.

---

> > > ### Author Response · Authors · 2024-08-09
> > >
> > > Thank you for your prompt response. We are pleased that we have addressed most of your questions. The remaining issue appears to stem from a misunderstanding, which we hope to clarify below. We hope that the reviewer will consider this in their final assessment.
> > >
> > > > **Question about novelty:** The reviewer acknowledges that the upper bound should be dependent on the environment's changes. [...] However, the statement "Our algorithm runs without needing knowledge of the environment changes as input" seems to imply a rather strong assumption. Does this mean that the algorithm requires exact information about the environment changes, or is it based on estimated information?
> > >
> > > **Answer:** There is still a misunderstanding. Our algorithm **does not** require prior-knowledge of the environment changes contrary to most existing works. The fact that it appears in the performance bounds is a strength, not a weakness of our result, and does not mean at all that the algorithm needs it to satisfy the bound.
> > >
> > > As the reviewer acknowledges, it is highly challenging to know the environment's changes with precision in a time-varying setting. Achieving our results without needing this knowledge was thus non-trivial and most existing algorithms loose their practicality contrary to us. This is one of our core contributions.
> > >
> > > Thanks again for pointing us to these related works. We will include them in our citations.
> > >
> > > > **Loop free terminology:** Could the authors please be more specific about the term "loop-free"?
> > >
> > > **Answer:** The loop-free problem refers to a specific case of episodic MDPs, where an agent must transverse episodes of a fixed length (denoted by $N$ in our case), always starting from the same state or from a state-action pair sampled from the same distribution (denoted by $\mu_0$ in our case), with transition probabilities dependent on the agent's time step $n \in [N]$. We will include these details in the main paper.

---

> ### Comment · Reviewer_ZTXg · 2024-08-10
>
> Thank you for the update.
>
> - **Novelty:** I apologize for the earlier confusion. I now understand what the authors mean. The algorithm does not require any prior information about the environment, and the performance bound is indeed dependent on the environmental changes. This point has been fully addressed.
>
> - **Loop-free:** Thank you for the explanation. This question is now fully addressed. It seems that "loop-free" refers to resetting the agent and starting from the initial stage multiple times. (Please correct me if I am mistaken.)
>
> I also appreciate how the author have written its. contribution in a explicit way and actively engaging in the  discussions. **I have raised my score to 7.** Please revise the paper based on our discussions if the paper get accepted. I will also consider the feedback from other reviewers. Good luck with your paper. Thank you.

---

> > ### Author Response · Authors · 2024-08-12
> >
> > Thank you for your quick response and for recognizing the new contributions of our paper. Yes, that is precisely what "loop-free" signifies in this context. We are pleased to have addressed all of the reviewer's questions, and we appreciate you taking the time to review our work.

---

### Author Rebuttal · Authors · 2024-08-06

We thank all the reviewers for their time and  feedback in evaluating our paper. While we agree that the assumption in the dynamics of Equation (8) may seem restrictive, we explain below why studying this case is important for CURL. Additionally, we highlight some of the novel contributions of our work.

- **New algorithm:** This is the first work where the framework of learning with expert advice has been adapted to online MDP to address non-stationarity. Previous RL approaches rely on sliding windows or restarts, which require prior knowledge of the variation budget. The first method to overcome this issue [44] does not handle adversarial losses, where losses can change arbitrarily at each episode and are unknown to the learner. Introducing the use of experts is novel for both RL and CURL, and can open the way for the development of new algorithms.

- **New analysis and need of dynamic assumption (8):**
    We emphasize that CURL (which is a convex objective) is significantly harder than RL (which is a linear objective). A notable example in the online learning literature is the comparison between convex bandits [Lattimore 2024] and multi-armed bandits [Lattimore and Szepesvari, 2020], where the former are highly more challenging. Early studies on more complex problems may need stronger assumptions to enable the development of new methods for general scenarios. Addressing online CURL needs a new theoretical analysis distinct from RL, even with the dynamics assumption in Equation (8). Existing work on online CURL, like [35], also requires this assumption and is limited to stationary environments.

    Our work adds the challenge of non-stationarity. In the meta-algorithm constructed using the experts framework, the complexity of working with CURL is addressed through our novel construction of $\hat{p}^t$, the estimator of the probability transition. This approach differs significantly from classic RL methods [Neu et al. 2012] and necessitates a new analysis (see Proposition 5.2).

- **Real world applications satisfying CURL with dynamic assumption (8):**
    Our setting encompasses many real-world problems outlined in the paper and further detailed below:
	- **Energy grid optimization.** To balance the energy production with the consumption, an energy provider may want to control the average consumption of electrical appliances (electric vehicles, water heaters, etc), with known physical dynamics but unpredictable and varying consumer behavior. This is a central problem in the context of climate change to introduce renewable energies into the electrical grid. The application paper Moreno et. al 2023 exactly fits into our CURL framework (see Eq. (1) of Moreno et al. 2023). Other related works are [41, Coffman et. al 2023].
	- **Robotics.** Controlling a population of drones often involves environments with non-stationary dynamics, such as changing weather conditions and human interventions. One objective might be to reach a certain target while avoiding specific states. This can be formulated as the following convex objective:
       $$F(\mu) := -\langle r, \mu \rangle + (\langle \mu, c \rangle)^2$$
	where $r$ is a reward vector and $c$ is a cost vector. Alternatively, the objective might be to distribute 	the drones uniformly throughout the space, which can be expressed using the entropy function:
	$$F(\mu) := \langle \mu, \log(\mu) \rangle.$$

	- **Inventory management.** Online resource allocation, where user demand is the unknown external noise independent of stock levels [Lee et al. 2023];

	- **Finance.** Trading tasks, assuming market independence from the trajectory of a single agent [Riva et. al 2022]; among others.

Since the external noise distribution and the cost function are unknown, we must still simultaneously learn the model and optimize the policy, which keeps us within the realm of reinforcement learning. These points provide enough motivation for addressing this setting, even though exploration is unnecessary.

### References
- *Lattimore 2024, Bandit Convex Optimisation*
- *Lattimore and Szepesvari 2020, Bandit algorithms*
- *Neu et. al 2012, The adversarial stochastic shortest path problem with unknown transition probabilities, AISTATS*
- *Moreno et. al 2023, (Online) Convex Optimization for Demand-Side Management: Application to Thermostatically Controlled Loads*
- *Coffman et. al 2023, A unified framework for coordination of thermostatically controlled loads, Automatica*
- *Lee et al. 2023, Online Resource Allocation in Episodic Markov Decision Processes*
- *Riva et al. 2022, Addressing Non-Stationarity in FX Trading with Online Model Selection of Offline RL Experts, ICAIF*

---

### Decision · Program_Chairs · 2024-09-25

**Decision:**

Accept (poster)

**Comment:**

This work explores online learning in non-stationary Markov decision processes (MDPs) where losses are non-linear and probability transitions are subject to change. The authors design an algorithm, MetaCURL, that runs multiple black box experts experts. Even though such an approach is relatively standard when studying dynamic regret, the presence of non-linearity together with time-changing transition model makes the setting challenging compared to prior work.

The main concerns reviewers raised are twofold: 1) the additional assumption on the transition model in equation 8 seems limiting, and 2) lack of motivating examples in light of lack of experiments. Regarding 1), the reviewers and I are still in agreement this assumption is restrictive, however, since the setting considered in this work is significantly more challenging there's a merit in making such restrictive assumption as a first step. Regarding 2), the authors elaborated on several examples that motivates the assumption in equation 8 and the setting studied in this work in general.

In light of these I am happy to accept this work. Further, I ask the authors to expand on the motivation in the camera ready version  so the community will benefit from these observations, as well as combine other suggestions from the reviewers.